# Robust derivation of transplantable dopamine neurons from human pluripotent stem cells by timed retinoic acid delivery

Zhanna Alekseenko [1], José M. Dias [1,10], Andrew F. Adler[2,3,10], Mariya Kozhevnikova[1,10], Josina Anna van Lunteren[4], Sara Nolbrant [2,3], Ashwini Jeggari[1], Svitlana Vasylovska [1], Takashi Yoshitake[5], Jan Kehr[5,6], Marie Carlén [4,7], Andrey Alexeyenko [8,9], Malin Parmar [2,3] & Johan Ericson [1✉]

Stem cell therapies for Parkinson's disease (PD) have entered first-in-human clinical trials using a set of technically related methods to produce mesencephalic dopamine (mDA) neurons from human pluripotent stem cells (hPSCs). Here, we outline an approach for high-yield derivation of mDA neurons that principally differs from alternative technologies by utilizing retinoic acid (RA) signaling, instead of WNT and FGF8 signaling, to specify mesencephalic fate. Unlike most morphogen signals, where precise concentration determines cell fate, it is the duration of RA exposure that is the key-parameter for mesencephalic specification. This concentration-insensitive patterning approach provides robustness and reduces the need for protocol-adjustments between hPSC-lines. RA-specified progenitors promptly differentiate into functional mDA neurons in vitro, and successfully engraft and relieve motor deficits after transplantation in a rat PD model. Our study provides a potential alternative route for cell therapy and disease modelling that due to its robustness could be particularly expedient when use of autologous- or immunologically matched cells is considered.

[1] Department of Cell and Molecular Biology, Karolinska Institutet, 171 65 Stockholm, Sweden. [2] Developmental and Regenerative Neurobiology, Department of Experimental Medical Science, Wallenberg Neuroscience Center, Lund University, 221 84 Lund, Sweden. [3] Lund Stem Cell Center, Lund University, 22184 Lund, Sweden. [4] Department of Biosciences and Nutrition, Karolinska Institutet, 141 83 Huddinge, Sweden. [5] Department of Physiology and Pharmacology, Karolinska Institutet, 171 65 Stockholm, Sweden. [6] Pronexus Analytical AB, Bromma, Sweden. [7] Department of Neuroscience, Karolinska Institutet, 171 65 Stockholm, Sweden. [8] Department of Microbiology, Tumor and Cell Biology, Karolinska Institutet, 171 65 Stockholm, Sweden. [9] Science for Life Laboratory, 171 21 Solna, Sweden. [10]These authors contributed equally: José M. Dias, Andrew F. Adler, Mariya Kozhevnikova. ✉email: johan.ericson@ki.se

Human pluripotent stem cells (hPSCs) in the form of embryonic stem cells (hESCs) or induced pluripotent stem cells (hiPSCs) provide a scalable cellular source for the production of specific subtypes of neurons that can be utilized for high-throughput drug discovery, disease modeling, or cell replacement therapy in neurodegenerative disorders[1,2]. Mesencephalic dopamine (mDA) neurons in the ventral midbrain (vMB) are of particular interest due to their relatively selective degeneration in Parkinson's disease (PD), and as pioneering clinical experiments using human fetal midbrain tissue have provided proof-of-concept that grafted dopamine neurons can restore dopamine neurotransmission and provide long-term relief in some PD patients[3].

Methods for in vitro derivation of human mDA neurons for clinical use have been progressively improved, and stem-cell based therapy for PD has, after extensive evaluation of safety and functional efficacy in preclinical animal models, entered the exciting stage of clinical trials using allogeneic hESCs or hiPSCs as starting material[2,4–6]. The use of allogeneic cells permit the production of large batches of frozen "off-the-shelf" preparations that can be used to treat many patients with the same cell product after validation for safety and efficacy in animal models, but a downside is a need for one year of immunosuppression to avoid rejection after transplantation[2]. Patient-specific hiPSCs are optimal from an immunological perspective, but add the significant challenge of establishing clinical-grade hiPSCs for each individual patient and to effectively direct these cells to adopt an mDA neuron fate prior to grafting. Patient-specific cells are therefore not likely to become a routine therapy at this stage of developments, albeit a recent case report provided proof-of-concept that an autologous approach for cell therapy is feasible[7]. The use of human leucocyte antigen (HLA)-matched hiPSC-lines is an alternative strategy to reduce immunosuppression but demands large numbers of selected HLA-typed hiPSC-lines to cover the majority of the population in a country[8]. Another interesting approach is to establish universal "one-fits-all" hPSC-lines that could provide immune tolerance without need for immunosuppression[9], but it is unclear when, and if, such lines will be available and approved for clinical use. Therefore, if restorative cell therapy is proven safe and effective in clinical trials, a likely development is that autologous and HLA-matched hiPSC-lines will be explored for routine clinical use. Apart from a standardized establishment of good manufacturing practice (GMP)-compliant hiPSC-lines, this will demand robust mDA neuron protocols that show low cell line- and batch variability in response to patterning agents in order to minimize laborious adjustments for individual cell lines.

Current state-of-the-art hPSC-based mDA neuron protocols utilize similar sets of developmental patterning signals to direct differentiation of hPSCs towards an mDA neuron fate. Dual SMAD inhibition (dSMADi) is applied to promote neural fate selection by preventing hPSCs from adopting alternative somatic or extraembryonic fate options via inhibition of TGF-beta and BMP signaling at early differentiation stages[10]. hPSC-derived neural progenitors adopt a forebrain (FB) fate in the absence of developmental patterning signals. Activation of canonical WNT signaling by the glycogen synthase kinase 3β (GSK3β) inhibitor CHIR90021 (CHIR), often applied together with FGF8, is used to impose a midbrain (MB) regional identity, by mimicking WNT1 and FGF8 signaling emanating from the isthmic organizer at the MB-hindbrain (HB) boundary[11]. Activation of the SHH pathway is in turn used to ventralize cells and induce LMX1A[+]/FOXA2[+]/ OTX2[+] ventral midbrain (vMB) floor plate progenitors, which can differentiate into functional mDA neurons after long-term culturing in vitro or after grafting into PD animal models[12,13]. Albeit the task of restoring dopamine neurotransmission and reversing motor deficits in preclinical models clearly can be

achieved using these methods, the anterior-posterior (AP) patterning response of differentiating hPSCs to CHIR is highly concentration-sensitive and requires careful protocol adjustments between cell lines[12,14–16]. Assessment of a large set of grafting experiments have further linked inter-experimental variability to imprecise vMB specification and contamination of diencephalic progenitors in spite of optimized CHIR-titration, which could be adjusted for by the complementing caudalizing activity of FGF8[17] or with biphasic CHIR-treatment (CHIR-boost)[18] which reduced diencephalic contamination and induced a more caudal EN1[+] vMB identity of cells. Nevertheless, a recent study establishes that ablation of 4 caudalizing patterning genes in hESCs promoting HB and spinal cord fates result in a more consistent and concentration-insensitive specification of vMB progenitors by CHIR, providing genetic support that CHIR-based patterning is inherently concentration-sensitive (preprint in bioRxiv)[19]. Development of methods that do not rely on CHIR to specify mesencephalic identity could therefore potentially result in more robust differentiation paradigms with lower batch-to-batch and cell line-to-cell line variability. Also, the high concentrations of CHIR used to impose midbrain identity are predicted to inhibit a broad array of kinases in addition to GSK3β[20], providing an additional incentive to consider differentiation strategies that do not rely on CHIR[21].

The isthmic organizer is a secondary signaling center established after the regionalization of the rostral neural plate into brain territories has been initiated[22]. Early brain patterning consequently involves signals operating upstream of WNT1 and FGF8, and certain observations implicate that the vitamin A-derivative Retinoic Acid (RA) may contribute to this process. The strong caudalizing activity of RA is well-established[11]. It is therefore commonly assumed that early RA exposure is incompatible with derivation of neurons with a more rostral origin in the central nervous system, and RA or vitamin A are often actively avoided in hPSC-based mDA neuron protocols[15,23,24]. However, studies have reported a caudal extension of FB and MB markers in vitamin A-deficient quail embryos[25] and there is a rostral expansion of the HB in mice lacking the RA-degrading enzymes CYP26A1 and CYP26C1[26]. Additionally, studies in mice have defined an early transient time window when FB tissue can be re-specified into a MB-like identity by RA[27]. This implies a potential to apply RA to impose midbrain-character to hPSC-derived NSCs. RA is also known to induce dissolution of pluripotency[28] and to promote maturation of naïve NSCs at initial stages of neural development[29], and could therefore possibly promote a rapid conversion of hPSCs into NSCs. In this study, we show that a 48-hour RA pulse in combination with SHH pathway activation promote a rapid specification of LMX1A[+]/FOXA2[+]/OTX2[+] vMB progenitors that differentiate into functional mDA neurons at high yield in vitro, and which engraft and restore motor deficits after transplantation into a rat model of PD. Finally, by simply tuning the duration of RA exposure, we could effectively generate serotonergic neurons, showing that RA-based patterning primarily relies on duration of signal exposure and provide proof-of-concept that this property has broad applicability and can be used to produce distinct clinically relevant neuronal subtypes from hPSCs.

## Results

**A 48-hour RA-pulse promotes rapid conversion of hPSCs into NSCs with midbrain-like identity.** To explore the activities of RA on hESCs directed to adopt a neural fate in response to dSMADi[10], we treated cultures of the GMP-compliant hESC line HS980 grown under dSMADi-conditions with 200 nM all-trans RA for the first 1, 2, 3, or 4 days of differentiation (RA[1D], RA[2D],

RA$^{3D}$, RA$^{4D}$) (Fig. 1a) and monitored fate and identity of cells at different stages by immunoblotting, quantitative immunocytochemistry, qPCR, or RNA-sequencing (RNA-seq). Consistent with previous studies[10], cells grown in dSMADi-only condition underwent a progressive transition from a pluripotency state (OCT4$^+$) into a naïve NSC state (SOX1$^+$ PAX6$^+$) between 0–7 days in differentiation condition (DDC) (Fig. 1b–d). OCT4 was gradually downregulated and approached undetectable levels at ~5 DDC, as revealed by quantitative immunocytochemistry (Fig.1c). Upregulation of SOX1 and PAX6 at low levels occurred at ~3 DDC (Fig. 1b). OCT4 and SOX1 were co-expressed by cells between 3–4 DDC (Fig. 1b, c and Supplementary Fig.1a) showing that the conversion of hPSCs into NSCs in response to dSMADi encompasses a notable time-window over which expression of pluripotency- and neural-specific genes overlap. In contrast, in cultures treated with dSMADi and 200 nM RA for two days or longer (dSMADi + RA$^{2D,3D}$), induction of SOX1 and PAX6 was observed at 2 DDC (Fig. 1e and Supplementary Fig. 1a) and expression of OCT4 had essentially been extinguished by 3 DDC (Fig. 1b, c, f). Expression levels of SOX1 and PAX6 at 3–4 DDC were notably higher in dSMADi+RA$^{2D}$-cultures relative to dSMADi-only cultures (Fig. 1b, c, f and Supplementary Fig. 1a). By 7 DDC, the expression of SOX1 and the neural stem cell marker NESTIN was similar in dSMADi-only and in dSMADi+RA$^{2D}$ cultures (Fig.1d). RNA-seq analysis of dSMADi+RA$^{2D}$-cultures suggested an overall downregulation of pluripotency genes and upregulation of neural lineage-specific genes at 2 DDC (Fig. 1e and Supplementary Table 1). Endodermal or mesodermal lineage markers were not upregulated (Supplementary Fig. 1b). Prompt suppression of OCT4 and fast upregulation of SOX1 was attained within an RA concentration-range between 50–500 nM when cells were grown in dSMADi+RA$^{2D}$ conditions (Fig. 1g). Treatment of cells with 200 nM RA for one day (dSMADi + RA$^{1D}$) was not sufficient to promote rapid OCT4 suppression or fast upregulation of SOX1 (Fig. 1f), nor was treatment of cells only with RA (without dSMADi) (Supplementary Fig. 1c). Accordingly, combining dSMADi with RA treatment for 48 h or longer promotes a rapid and switch-like transition from a pluripotency state into a NSC state.

To determine the regional identity of hPSC-derived NSCs exposed to RA for different timeframes, we analyzed the expression of transcription factors whose expression alone or in combination distinguishes between FB, MB, or HB regional identities in cultures at 9 DDC. dSMADi treatment was included in all following experiments and will not be further highlighted when describing experimental setups. As expected, in hPSC-cultures not treated with RA (RA$^{0D}$), NSCs acquired a FOXG1$^+$/OTX2$^+$/HOXA2$^-$ FB-like identity (Fig. 1h). A similar FB-like character was observed in RA$^{1D}$ cultures, though the level of FOXG1$^+$ expression was somewhat reduced (Fig. 1h and Supplementary Fig. 1d). Interestingly, in RA$^{2D}$ cultures, FB markers were suppressed and NSCs instead expressed a FOXG1$^-$/OTX2$^+$/HOXA2$^-$ MB-like character (Fig. 1h and Supplementary Fig. 1d). In RA$^{3D}$ and RA$^{4D}$ cultures, NSCs acquired a FOXG1$^-$/OTX2$^-$/HOXA2$^+$/HOXB4$^-$ rostral HB and FOXG1$^-$/OTX2$^-$/HOXA2$^+$/HOXB4$^+$ caudal HB identities, respectively (Fig. 1h). These data show that a 48-hour RA-pulse suppresses FB fate and imposes a MB-like identity to hPSC-derived NSCs, while longer exposure results in HB identity (Fig.1i).

**Combined RA and SHH signaling imposes a ventral midbrain identity to hPSC-derived NSCs.** We next activated SHH signaling to impose a ventral identity to NSCs by treating cultures with the Smoothened agonist SAG[30] between 0–9 DDC, and analyzed the fate of cells exposed to RA for different timeframes by immunocytochemistry or RNA-seq (Fig. 2a). At 9 DDC, NSCs

generated in SAG-only or RA$^{1D}$ + SAG conditions expressed the FB-specific markers FOXG1, SIX3, SIX6, and LHX2 (Fig. 2b) and the ventral marker NKX2.1 (Fig. 2a, b) which is a selective marker for the ventral telencephalon and diencephalon[31,32]. In RA$^{2D}$ + SAG cultures, FB markers were suppressed and NSCs adopted a LMX1A$^+$/LMX1B$^+$/FOXA2$^+$/OTX2$^+$ identity characteristic of vMB mDA neuron progenitors (Fig. 2a, b, d). In RA$^{4D}$ + SAG cultures, cells expressed HOX genes and the ventral markers NKX2.2, PHOX2B, NKX6.1, and NKX6.2 typical of cranial motor neuron (MN) progenitors of the HB[33] (Fig. 2a, b, d). Principal component and hierarchal clustering analyses of RNA-seq data showed clear segregation and broad transcriptional changes between cells exposed to RA for different timeframes (Fig. 2c and Supplementary Fig. 2a). Collectively, these data establish first that stepwise increases in duration of RA exposure imposes progressively more caudal regional brain identities (FB- > MB- > HB) to hPSC-derived NSCs (Fig. 1i). Second, when combined with SHH pathway activation, a 48 h RA-pulse appears sufficient to impose a LMX1A$^+$/FOXA2$^+$/OTX2$^+$ vMB-like identity to NSCs. Effective specification of a LMX1A$^+$/FOXA2$^+$ vMB-like identity was attained when the 48 hour RA-pulse was initiated between 0–2 DDC (Supplementary Fig.2b) and when SAG treatment was initiated at 0 or 1 DDC (Supplementary Fig.2c) at a concentration ≥50 nM (Supplementary Fig.2d).

A LMX1A$^+$/LMX1B$^+$/FOXA2$^+$/OTX2$^+$ identity was long considered as a molecular hallmark for vMB progenitors generating mDA neurons, but this identity was shown to be shared also by ventral progenitors of the caudal diencephalon giving rise to subthalamic nucleus (STN) neurons[17,32] (Fig. 2d). BARHL1, BARHL2, PITX2, and NKX2.1 are selectively expressed by the STN-lineage and thus can be used to distinguish between diencephalic STN-progenitors and vMB progenitors[32]. Analyses of RA$^{2D}$ + SAG cultures after 14 days showed that the vast majority of LMX1A$^+$ cells co-expressed FOXA2, OTX2, and LMX1B as well as the vMB marker CORIN[34] (Fig. 2g; LMX1A$^+$FOXA2$^+$: 92.5% ± 3.3, OTX2$^+$: 97.3% ± 1.7, mean ± SD, $n = 4$). At this stage, a minor subset of cells had initiated expression of NURR1 (Fig. 2g), an early marker of post-mitotic mDA neurons[35]. RNA-seq data showed negligible expression of BARHL1, BARHL2, PITX2, and NKX2.1 (Fig. 2e) and rare LMX1A$^+$ or LMX1B$^+$ NSCs co-expressed NKX2.1, PITX2, or BARHL1 (Fig. 2g; NKX2.1$^+$: 2.1% ± 2 ($n = 3$), PITX2$^+$: 1.4 ± 0.8 ($n = 2$); mean ± SD) in 14 DDC cultures. Expression of NKX2.2, PHOX2B, PHOX2A, and NKX6.1 either alone or in combination define progenitors giving rise to cranial motor neurons (MNs) and serotonergic neurons (5HTNs) in the ventral HB[33,36], or oculomotor neurons[37] and GABAergic neurons[38] derived lateral to mDA neurons in the MB (Fig. 2d). RNA-seq data revealed low expression of these markers in RA$^{2D}$ + SAG cultures (Fig. 2e) and few cells expressed NKX2.2, PHOX2A, PHOX2B, and NKX6.1 at 14 DDC as determined by immunocytochemistry (Fig. 2g). qPCR analyses of six biological replicates of RA$^{2D}$ + SAG cultures isolated at 14 DDC indicated consistent vMB specification and low expression of inappropriate regional markers (Fig. 2h).

WNT1 and FGF8 signaling emanating from the isthmic organizer impose a caudal-high to rostral-low gradient of EN1 and EN2 expression in the MB[39,40]. EN1, EN2, WNT1, FGF8 and the isthmic markers PAX2, PAX5, and PAX8 were expressed at very low or undetectable levels at 14 DDC (Fig. 2e). Activation of canonical WNT signaling by CHIR99021 is associated with translocation of β-catenin into nuclei[12,21] (Fig. 2f). There was no nuclear accumulation of β-catenin in response to RA treatment (Fig. 2f) and no induction of WNT1 or the WNT response gene AXIN2, nor FGF8 in response to RA in ESC-cultures at 2 DDC to 14 DDC (Fig. 2e and Supplementary Fig. 2e) Together, this suggest that LMX1A$^+$/FOXA2$^+$/OTX2$^+$ cells induced by RA and

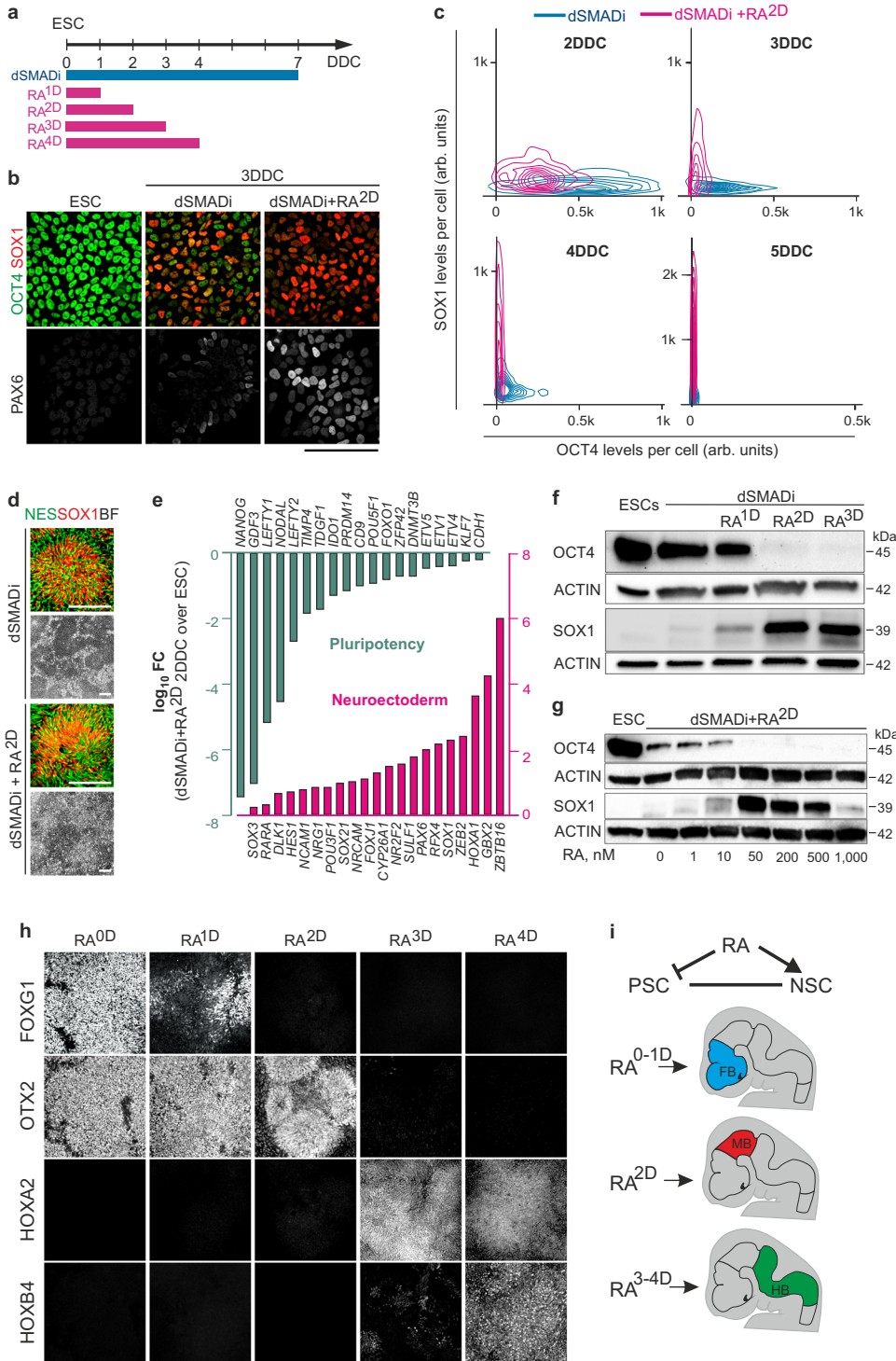

**Fig. 1 A two-day RA-pulse results in a rapid induction of NSCs expressing a MB-like identity. a** Schematic of hPSC differentiation with timeline of treatment with dSMADi and RA. **b** Immunocytochemistry for the pluripotency marker OCT4 and neuroectodermal markers SOX1 and PAX6 on ESCs and 3-day cultures differentiated in dual SMAD inhibitors (dSMADi) or in dSMADi with a two-day RA-pulse (dSMADi + RA2D). **c** Density plot of single cell expression of SOX1 and OCT4 in cultures differentiated in dSMADi or dSMADi+RA2D at indicated days in differentiation conditions (DDC). 670-760 cells per condition and time point were quantified in a representative differentiation. **d** Immunocytochemistry for neural progenitor markers NES and SOX1, and bright-field (BF) images of cultures at 7 DDC differentiated in the indicated conditions. **e** RNAseq-based $log_{10}$ fold change values of expression of genes associated with pluripotency or neuroectodermal fate in dSMADi+RA2D cultures at 2 DDC relative to ESCs. P value and FDR in Supplementary Table 1, $n = 2$ independent experiments. **f** Representative western blot for OCT4 and SOX1 of ESC and 3 DDC cultures differentiated in indicated conditions. **g** Representative western blot for OCT4 and SOX1 of ESC and 3 DDC cultures grown in dSMADi-conditions and treated with indicated concentrations of RA for two days. **h** Immunocytochemistry for markers identifying forebrain (FOXG1), forebrain and midbrain (OTX2), hindbrain (HOXA2), and caudal hindbrain (HOXB4) regions in nine DDC cultures differentiated in dSMADi and treated with RA as indicated. **i** Summary of the effects of RA-pulse duration on NSC's regional identity. Scale bars, 100 µm. Arb.units arbitrary units, FB forebrain, MB midbrain, HB hindbrain.

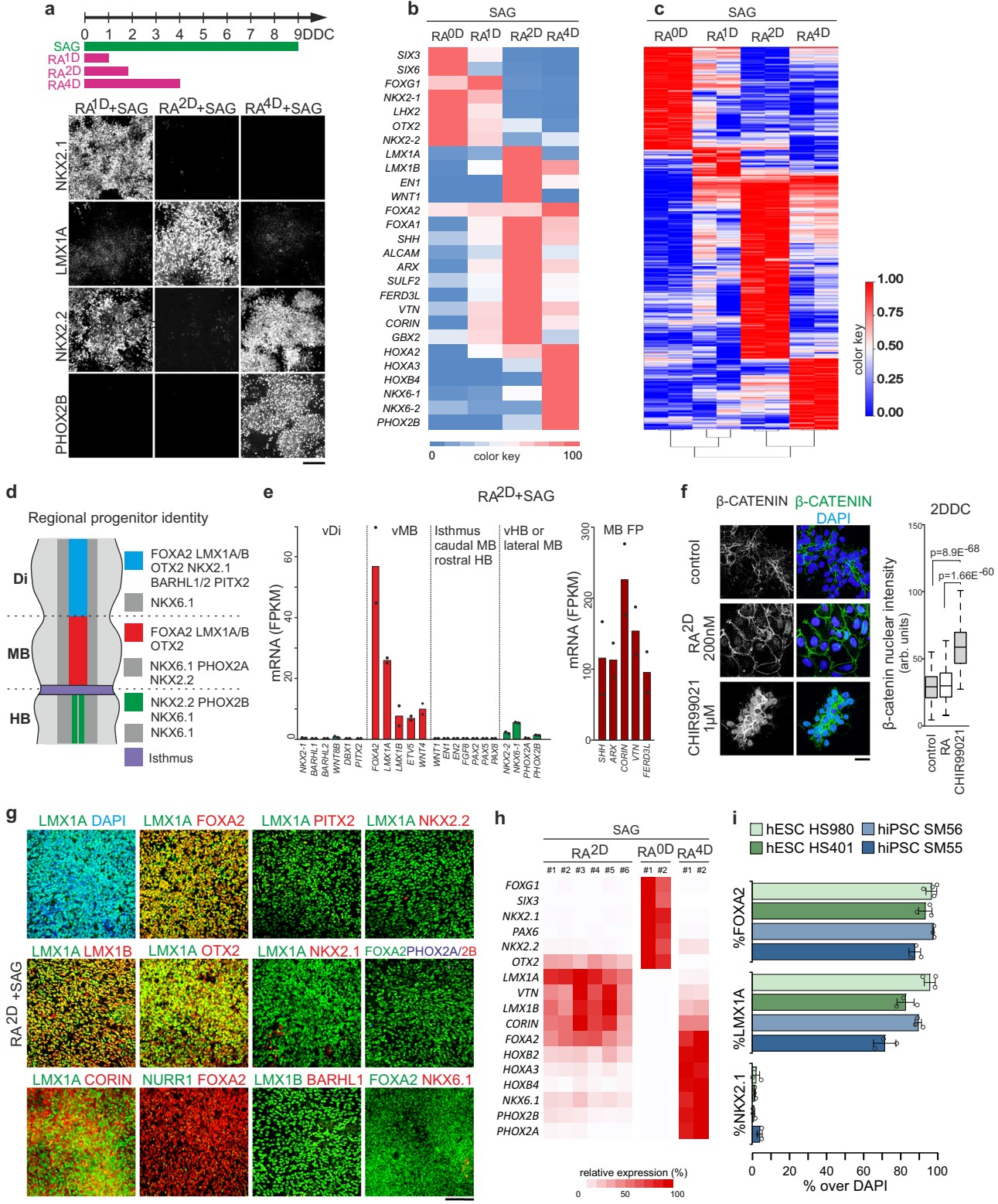

SAG acquire a EN1$^{-/LOW}$ rostral-like vMB identity and that specification of vMB-fate can occur independently of WNT1, FGF8 or induction of isthmic organizer-like cells in hPSC-cultures.

Next we compared the patterning response to RA$^{2D}$ + SAG treatment for one additional hESC-line HS401 as well as two hiPSC-lines SM55 and SM56. For each cell line analyzed at 14 DDC, we observed consistent induction of LMX1A$^{+}$/FOXA2$^{+}$/ OTX2$^{+}$/NKX2.1$^{-}$ vMB progenitors at high yield without need to adjust concentration or time of RA exposure (Fig. 2i and Supplementary Fig. 3), which would normally require re-titering of patterning agents using other mDA neuron protocols where CHIR is used for caudalization[15]. Collectively, these data suggest that RA-based differentiation promotes robust and reproducible vMB specification with high inter-experimental consistency and low cell line variability.

**Fig. 2 Specification of NSCs with ventral midbrain identity using RA and SHH signaling. a** Schematic of hPSC differentiation with timeline of treatment with RA and SHH signaling activator (SAG) (top). All cultures were differentiated in dSMADi conditions as indicated in Fig. 1a. Immunocytochemistry for indicated markers at 9 DDC in cultures differentiated in +SAG-condition and pulsed with RA for 1-, 2-, or 4-days (bottom). **b** Normalized gene expression of indicated genes in 9 DDC cultures differentiated in +SAG and treated with RA for the indicated time ($n = 2$ independent experiments per condition). **c** Heatmap and hierarchical clustering of differentially expressed genes of 9 DDC cultures differentiated in +SAG and treated with RA for the indicated time ($n = 2$ independent experiments per condition). **d** Schematic of gene expression profiles defining distinct ventral progenitors in the diencephalon (Di), midbrain (MB), and hindbrain (HB). **e** Expression of genes associated with indicated regional progenitors at 14 DDC in cultures differentiated in +SAG + RA$^{2D}$ in two independent experiments. **f** Representative immunocytochemistry for β-CATENIN (left) and boxplots of β-CATENIN nuclear levels (right) in 2 DDC cultures differentiated in +SAG (control) and treated with RA or CHIR99021. Boxplot data: center line defines median, the box indicates quartile 1 to quartile 3, and whiskers indicate ±1.5x interquartile range. $n = 227$ (control), 247 (RA), 232 (CHIR99021) cells examined over 3 independent differentiations, $p$ values derived from unpaired two-tailed $t$-test. **g** Immunocytochemistry for the indicated markers in cultures at 14 DDC differentiated in dSMADi+SAG + RA$^{2D}$. **h** qPCR analysis of a panel of 17 genes for six independent mDA differentiations at 14 DDC (RA$^{2D}$ + SAG column; differentiations #1-#6). mDA samples were compared to cultures differentiated to a forebrain (RA$^{0D}$ + SAG, $n = 2$) or hindbrain (RA$^{4D}$ + SAG, $n = 2$) progenitor identity. **i** Quantification of the percentage of cells expressing FOXA2, LMX1A or NKX2.1 in 14 DDC cultures differentiated in +RA$^{2D}$ + SAG across four different hPSC lines (values, mean ± SD, $n = 3$–4). Scale bars: 100 μm in panels (**a**, **g**), 25 μm panel (**f**).

**Self-enhanced RA degradation via CYP26A1 mediates robust vMB patterning.** To understand the robust patterning activity of RA, we examined patterning output in response to altered concentrations of RA. In these experiments, cells cultured in RA$^{2D}$ + SAG-conditions were exposed to RA in the range of 100–800 nM, and the regional identity of cells was analyzed at 9 DDC. In cultures exposed to 200-400 nM RA, the vast majority of NSCs expressed a LMX1A$^+$/NKX2.1$^-$ vMB-identity and few cells expressed a diencephalic LMX1A$^+$/NKX2.1$^+$ identity or NKX2.2$^+$/LMX1A$^-$ HB-like identity (Fig. 3a, c). It is notable that some LMX1A$^+$ at 9 DDC co-expressed NKX2.2 at very low levels (Supplementary Fig. 2c), which is consistent with data showing that NKX2.2 is transiently expressed in vMB progenitors at early developmental stages in vivo[37]. LMX1A$^+$/NKX2.1$^-$ vMB progenitors were generated also when the RA concentration was reduced to 100 nM or increased to 800 nM RA, but at a lower yield (Fig. 3c). When we used CHIR to specify vMB identity, cells attained a LMX1A$^+$/NKX2.1$^-$ vMB identity in response to 1 μM CHIR, but the regional identity of NSCs shifted into an anterior diencephalic LMX1A$^+$/NKX2.1$^+$ character when the concentration was reduced to 0.8 and 0.6 μM (Fig. 3b, c) or into a posterior NKX2.2$^+$/LMX1A$^-$ presumptive HB identity when the concentration was raised to 1.2 and 1.4 μM (Fig. 3b, c). These data demonstrate that specification of vMB regional identity is relatively tolerant to altered RA concentrations and that it is instead the duration of RA exposure that is central for robust vMB patterning.

The *CYP26* family of genes (*CYP26A1, CYP26B1, CYP26C1*) encode enzymes of the cytochrome p450 family that metabolize RA through oxidation[41]. CYP26A1 is expressed by the rostral-most neuroectoderm and contributes to prevent a rostral extension of HB identity at early stages of neural development[26]. Also, in anterior-posterior patterning of the HB, negative feedback regulation of RA signaling by self-enhanced degradation via induction of CYP26 proteins is central for shaping RA gradients and to buffer for fluctuations of RA levels[42,43]. We observed a RA concentration-dependent activation and adaptive temporal expression profile of *CYP26A1* in hPSC cultures (Fig. 3d, e). This provides a plausible mechanistic rationale for the robust patterning response of hPSCs to RA, as altered RA input can be buffered by a matching change of the rate of RA turnover by CYP26A1. To explore this, we examined the patterning output in response to timed RA exposure after inhibiting CYP26 activity with the selective antagonist R115866[44] between 0 and 3 DDC. In RA$^{1D}$ + SAG or RA$^{2D}$ + SAG cultures treated with 500 nM R115866, cells acquired a PHOX2B$^+$ HB-identity instead of a NKX2.1$^+$ FB-identity or LMX1A$^+$/NKX2.1$^-$ vMB-identity at 9 DDC, respectively (Fig. 3f). *HOXA2* and *HOXB4* were induced in these experiments suggesting a caudal HB identity (Fig. 3g),

which corresponds to a regional identity acquired after four days of RA exposure if CYP26 function is left intact (Fig. 1h). When the R115866 concentration was reduced to 100 nM, FB fate was suppressed but cultures contained a mix of LMX1A$^+$/NKX2.1$^-$ vMB cells and PHOX2B$^+$ HB cells (Fig. 3g and Supplementary Fig. 4a), presumably reflecting that a partial inhibition of CYP26A1 produces an intermediate caudalizing effect. Importantly, treatment of cells only with R115866 and SAG did not suppress NKX2.1$^+$ FB-fate (Fig. 3f), establishing that the strong caudalizing effect of R115866 is RA-dependent.

Like all-trans RA, 9-*cis* RA, 13-*cis* RA, and the xenobiotic RA-analogue tazarotenic acid (TA) are substrates for CYP26-mediated oxidation[45,46]. Exposure of cells to 500 nM of these analogues for 48-hours mimicked the patterning activity of all-*trans* RA by imposing a LMX1A$^+$/NKX2.1$^-$ vMB identity (Fig. 3h and Supplementary Fig. 4b), and inhibition of CYP26 activity resulted in a shift into a PHOX2B$^+$ HB identity (Fig. 3h). The synthetic RA analogue EC23 is predicted to be resistant to CYP26 mediated oxidation[47] and when all-trans-RA was replaced with 200 nM of EC23, cells grown either in EC23$^{1D}$ + SAG or EC23$^{2D}$ + SAG conditions adopted a PHOX2B$^+$ HB-identity both with or without inhibition of CYP26 (Fig. 3i and Supplementary Fig. 4a). Titration experiments showed that EC23 could induce LMX1A$^+$/NKX2.1$^-$ vMB cells, but this was imprecise and required a 20-fold reduction in concentration and treatment of cells only for 24 h (Supplementary Fig. 4c). Together, these data establish that the AP-patterning output in response to timed RA exposure is critically reliant on the RA concentration-dependent activation of CYP26A1 in responding hPSCs, and provides a mechanistic rationale to explain robustness and tolerance to altered RA input in RA-mediated vMB specification (Fig. 3j).

**Effective generation of functional mDA neurons at high yield in vitro.** A unique feature of mDA neurons is that they originate from initially non-neuronal floor plate (FP) cells and progenitors must therefore acquire neuronal potential prior to differentiation into neurons[34,48]. Few markers distinguish between these vMB maturation stages, but downregulation of SHH and upregulation of pro-neural bHLH proteins over time correlate with this transition[48]. RNA-seq analyses of RA$^{2D}$ + SAG treated cells isolated at 9, 12, 14, and 21 DDC showed that the expression of pan-FP markers *SHH, CORIN, ARX, VTN, FERD3L, NTN1, SLIT2, SULF2*, and *ALCAM* peaked at around 12-14 DDC and subsequently declined (Fig. 4a, Supplementary Fig. 5a and Supplementary Table 2). Conversely, *NEUROG2, NEUROG1, NEUROD4*, and *ASCL1* encoding pro-neural bHLH proteins were upregulated at 21 DDC (Fig. 4a) as well as the mDA neuron markers *NR4A2* (*NURR1*) and *TH*, and the

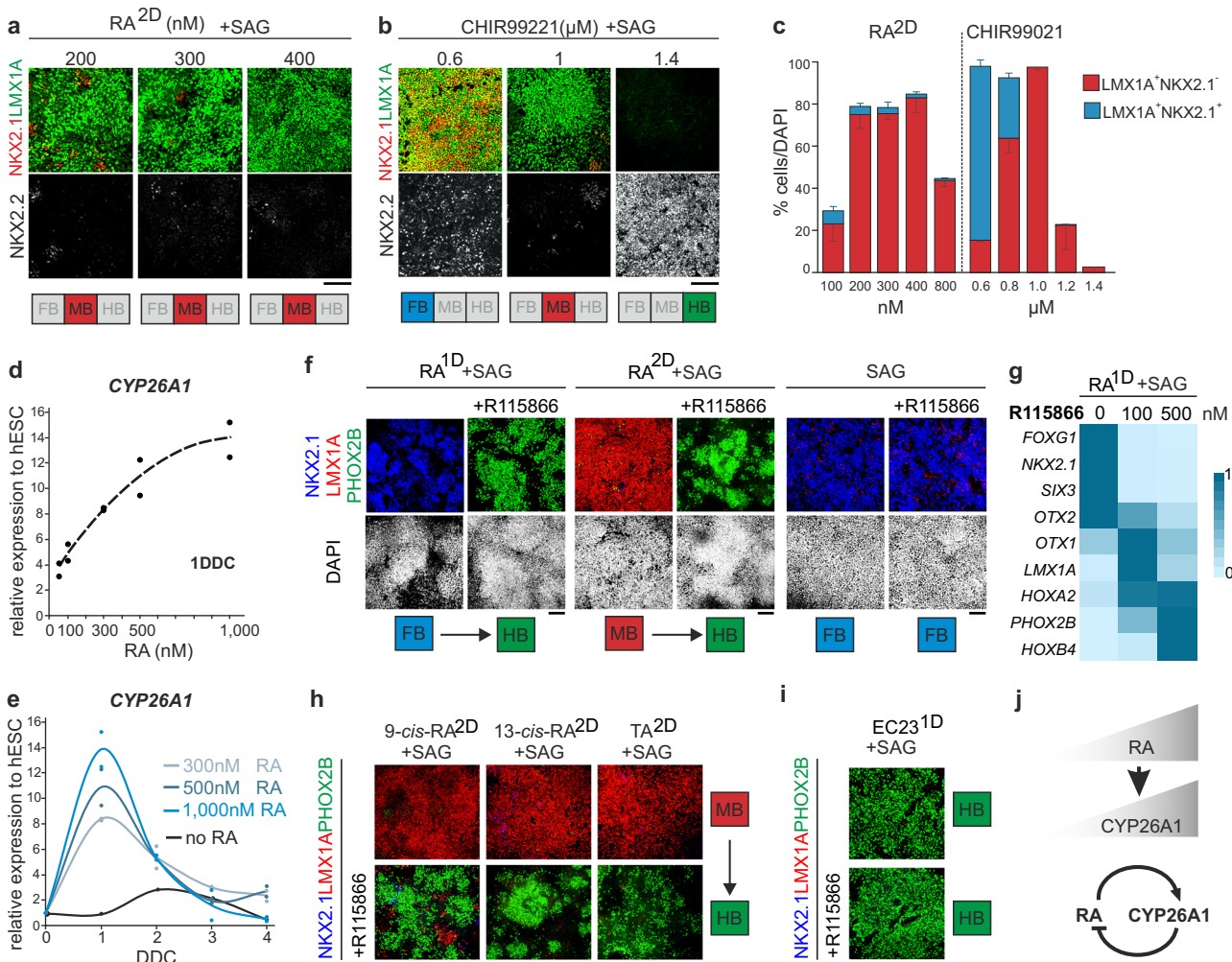

**Fig. 3 A RA-CYP26 regulatory loop is central for robust RA-mediated patterning response. a–c** Effect of changes in RA (**a**) or CHIR99021 (**b**) concentration on the expression of markers defining ventral forebrain (NKX2.1+LMX1A+), midbrain (NKX2.1−LMX1A+) or hindbrain (NKX2.2+) in 9 DDC cultures differentiated in dSMADi+SAG, and corresponding quantification of NKX2.1+LMX1A+ and NKX2.1−LMX1A+ populations ($n = 3$ independent differentiations per condition). values, mean ± S.D (**c**). **d** Effect of RA concentration (50, 100, 300, 500, and 1000 nM RA) on *CYP26A1* expression in 1 DDC cultures differentiated in dSMADi (data from two independent differentiations). **e** Temporal expression profile of *CYP26A1* in cultures differentiated in dSMADi and pulsed with no RA or 300, 500, or 1000 nM of RA for 2 days (data from two independent differentiations). **f** Effect of inhibition of CYP26 activity with 500 nM R115866 inhibitor on the expression of NKX2.1, LMX1A, and PHOX2B at 9 DDC in cultures differentiated in SAG and indicated RA conditions. **g** Relative gene expression of indicated genes in 9 DDC cultures differentiated in RA1D + SAG condition and in the presence of different concentrations of the CYP26 inhibitor R115866 (data from two independent differentiations). **h, i** Immunocytochemistry of NKX2.1, LMX1A, and PHOX2B at 9 DDC in cultures differentiated in dSMADi plus the indicated conditions (500 nM 9-cis-RA, 13-cis-RA and TA; 200 nM EC23) and in the presence or absence of 500 nM R115866. **j** Schematic summary of the regulatory interactions between RA and CYP26A1. Scale bars 100 μm.

pan-neuronal markers *DCX*, *TUBB3*, and *STMN2* (Fig. 4a and Supplementary Table 2). Immunocytochemical analyses confirmed that SHH expression was higher at 14 DDC as compared to 21 DDC and that the number of NEUROG2+ progenitors increased over this period (Fig. 4b). There was a progressive accumulation of HuC/D+ neurons between 14 and 21 DDC (Fig. 4c, d), and by 21 DDC, ~30% of DAPI+ cells accounted for HuC/D+ neurons in RA2D + SAG cultures (Fig. 4d). Many of these had also initiated expression of TH (Fig. 4c). When we applied CHIR + SAG to specify LMX1A+/ NKX2.1− vMB progenitors (Fig. 3b) cells initiated neurogenesis at around 17 DDC (Fig. 4c, d) which is consistent with previous studies[13,15]. At 21 DDC, HuC/D+ neurons constituted ~10% of total cells and few neurons expressed TH (Fig. 4c, d). Similar results were observed when CHIR + SAG-treated cultures were complemented with FGF8b-treatment between 9–16 DDC[15,17] (Fig. 4c). This indicate that RA-specified vMB progenitors initiate neurogenesis early and that cells at the population level undergo relatively coordinated

conversion from an immature FP state to a neurogenic mDA neuron progenitor state.

RA-specified TH+ neurons acquired a progressively more advanced neuronal morphology with progressive outgrowth and branching of axonal processes between 30–45 DDC (Fig. 4e, j, k). TH+ neurons expressed mDA neuron markers LMX1B, LMX1A, FOXA2, NURR1, and OTX2 at 30–35 DDC (Fig. 4 e–g). Only rare cells expressed Ki67 or phospho-histone H3 at 35–40 DDC (Supplementary Fig. 5b) indicating low mitotic activity in long-term cultures. ~80% of all neurons expressed TH+ at 35 DDC (Fig. 4h, i). ~10% of neurons expressed GABA (Fig. 4h, i), and some of these co-expressed TH (Supplementary Fig. 5b). Only rare neurons expressing 5-HT or the MN marker PRPH were detected (Fig. 4h, i). All four hPSC-lines examined in this study were confirmed to give rise to a high content of TH+ neurons (~60–80% of total neurons) in long-term cultures (Supplementary Fig. 5c).

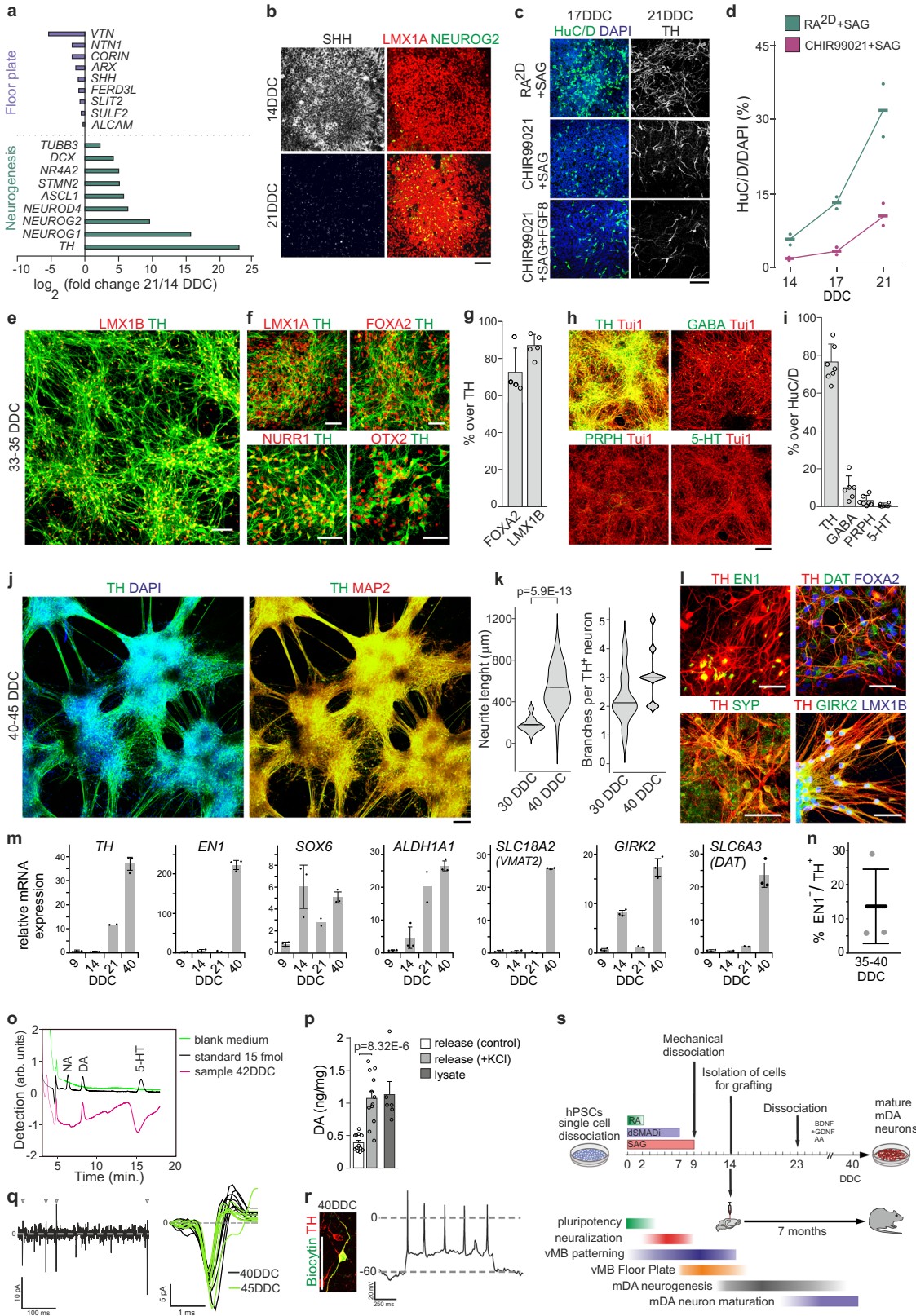

Temporal mRNA expression analyses revealed a notable upregulation of *EN1* in ESC-cultures at 40 DDC (Fig. 4m) and a small fraction of TH$^+$ neurons (~15%) expressed detectable EN1 based on immunocytochemistry (Fig. 4l, n). Accordingly, while immature RA-specified rostral-like vMB progenitors show negligible *EN1* expression (Fig. 4m), this indicates that EN1 becomes upregulated in differentiating mDA neurons over time (see also below). Cells also expressed the mDA neuron maturation markers DAT, GIRK2, SYNAPTOPHYSIN, CALB1, *SOX6, VMAT2* and *ALDH1A1* at 40 DDC, as determined by immunocytochemistry (Fig. 4l and Supplementary Fig. 5b, d) or mRNA expression (Fig. 4m). High performance liquid

**Fig. 4 Generation of functional dopaminergic neurons in vitro. a–r** Analysis of cultures differentiated in dSMADi+RA$^{2D}$ + SAG (**a**, **b**, **e–r**) or dSMADi+SAG and indicated RA, CHIR99021, or CHIR99021 + FGF8 condition (**c**, **d**). **a** RNAseq-based $log_2$ fold change values of expression of genes associated with floorplate identity and neurogenesis at 21 DDC relative to 14 DDC. P- and FDR-values in Supplementary Table 2, data from two independent experiments ($n = 2$). **b** SHH, LMX1A, and NEUROG2 immunocytochemistry at 14 and 21 DDC. **c** Immunocytochemistry of neuronal marker HuC/D at 17 DDC and dopaminergic neuron marker TH at 21 DDC in indicated conditions. **d** Quantification of HuC/D$^+$ neurons at 14, 17, and 21 DDC in indicated conditions ($n = 2$). **e–i** Immunocytochemistry of indicated markers (**e**, **f**, **h**) and quantifications of TH$^+$ cells expressing FOXA2 or LMX1B ($n = 4$ and $n = 5$, respectively) (**g**), and of dopaminergic (TH$^+$, $n = 7$), GABAergic (GABA$^+$, $n = 6$), motor (PRPH$^+$, $n = 7$), and serotonergic (5-HT$^+$, $n = 7$) neurons at 33–35 DDC. **i** Values, mean ± S.D.TH/Tuj and 5-HT/Tuj images in (**h**) correspond to a triple 5-HT/TH/Tuj immunocytochemistry. **j, l** Immunocytochemistry of indicated markers in 40–45 DDC cultures. **k** Violin plots of neurite length (center line, mean; $p$ value from pairwise Wilcoxon test) and of neurite branching quantification (center line, mean) in 30 and 40 DDC TH$^+$ neurons, $n = 40$ neurons per time point. **m** qPCR of mDA markers during mDA differentiation ($n = 3$, except for 21 DDC where $n = 2$). **n** Quantification of TH$^+$ cells expressing EN1 at 35–40 DDC ($n = 3$). Data in (m,n) as mean ± S.D. **o** HPLC detection of noradrenaline (NA), dopamine (DA), and serotonin (5-HT) in the supernatant of 42 DDC cultures. **p** Quantification of dopamine levels in media at 42 DDC in control ($n = 12$) or after KCL induced dopamine release ($n = 12$), and in total cell lysate ($n = 6$). Values presented as mean ± SEM, $p$ value from two-tailed unpaired $t$-test. **q** Cell-attached recording, showing spontaneous action potentials (left) and isolated spontaneous action potentials (right) from cells at 40–45 DDC. Gray arrowheads denote spontaneous spikes. **r** Immunocytochemistry for TH on Biocytin labelled neuron (left) and evoked spike train (right) in 40 DDC neuron. Schematic summary of differentiation timeline. Scale bars: 100 μm in (**b**, **c**, **e**, **h**, **j**) and 50 μm (**f**, **l**, **r**).

chromatography (HPLC) analyses at 42 DDC established that neurons produced and released dopamine (Fig. 4o, p), but not serotonin (5-HT) or noradrenaline (NA) (Fig. 4o). The time hPSC-derived cells acquire spontaneous action potentials, evoked action potentials and voltage-dependent Na$^+$ and K$^+$ currents provide another measure to evaluate the maturation state of mDA neurons in vitro[49]. In RA$^{2D}$ + SAG-cultures, the number of patched cells showing voltage-dependent Na$^+$ and K$^+$ currents was low at 35 DDC but increased notably at 38 DDC and remained at a largely constant level thereafter (Supplementary Fig. 6), and neurons exhibiting spontaneous (Fig. 4q) or evoked action potentials (Fig. 4r) were recorded at 40–45 DDC. Collectively, these data establish that RA-specified vMB progenitors differentiate into functional mDA neurons at high yield in vitro (Fig. 4s), with little contamination of neuronal subtypes generated in close proximity to mDA neurons in the developing brainstem.

**RA-specified cells engraft and reverse motor deficits in a rat model of PD.** To determine the in vivo performance of vMB progenitors specified in response to RA$^{2D}$ + SAG, we transplanted vMB preparations in a pre-clinical rat model of PD[50]. vMB progenitors were isolated at 14 DDC (Fig. 4s) and grafted to the striatum of athymic (nude) rats with prior unilateral 6-OHDA lesion to the medial forebrain bundle as previously described[17] (Supplementary Fig. 7a). Seven months after transplantation, immunohistochemistry analysis showed that all nine rats had surviving, neuron rich grafts with extensive innervation of DA target regions including dorsolateral striatum and prefrontal cortex (Fig. 5a). The grafts contained a large number of TH$^+$ neurons (4300 ± 47 TH$^+$ neurons per graft, Fig. 5b) which co-labeled with the human marker HuNu (Fig. 5c, d and Supplementary Fig. 7b). Graft-derived TH$^+$ neurons also co-expressed FOXA2, LMX1A/B, PITX3, NURR1, and EN1 (Fig. 5e–j), indicating that they adopted an mDA phenotype also after transplantation. ~70% of TH$^+$ neurons expressed detectable EN1 after grafting, showing that a majority of RA-specified mDA neurons expressed EN1 after differentiation and maturation in vivo (Fig. 5n). ≥70% of TH$^+$ neurons also co-expressed PITX3 as well as GIRK2 and SOX6, which are markers enriched in therapeutic A9-subtype of mDA neurons[51] (Fig. 5i–l, n). The A9 identity was further supported by TH$^+$ fibers innervating the surrounding dorsolateral striatum (Fig. 5a and Supplementary Fig. 7c). Innervation was also detected in prefrontal cortex and a small fraction of TH$^+$ neurons expressed CALB1 (20.5% ± 4.7, mean ± SD) (Fig. 5m, n), a marker enriched in A10 subtypes of mDA neurons[51]. The functionality of the TH$^+$ neurons was assessed using drug induced and spontaneous motor tests; amphetamine-induced rotation and stepping test, which demonstrated functional recovery 7 months post transplantation (Fig. 5o–q). Together, these results show that hPSC-derived vMB progenitors specified by RA and SAG successfully engraft, extend projections to relevant target areas, and differentiate into functional mDA neurons that restore motor deficits in an animal model of PD to a degree comparable to what has been reported from cells generated using CHIR-based patterning protocols[13,17,52,53].

**Timed RA exposure to specify distinct types of clinically relevant neurons.** While our study focuses on RA-based specification of mDA neurons, timed RA exposure should be broadly applicable for specification of various types of clinically relevant neurons from hPSCs. For instance, prolonged RA exposure (≥ 3 days) together with SHH pathway activation induces NKX2.2$^+$ HB-like progenitors (Fig. 2a, b) which are known to sequentially generate cranial PHOX2B$^+$ motor neurons (MN) and serotonergic neurons (5HTNs) during mouse and human development[36,54]. Since 5HTNs modulate a broad array of physiological processes and behaviors, and are implicated in the pathophysiology of a spectrum of psychiatric conditions[55,56], we examined if timed RA treatment could provide a strategy also for the production of human 5HTNs[16,55] (Fig. 6a). The time-window over which MNs are produced is defined by the co-expression of NKX2.2 and the MN-determinant PHOX2B in progenitors[57]. In RA$^{4D}$ + SAG-cultures, NKX2.2 and PHOX2B were co-expressed between ~9–16 DDC (Fig. 6b) and there was a progressive accumulation of differentiating MNs over this period, as determined by the presence of PHOX2B$^+$/NKX2.2$^-$ cells at 17 DDC, and PHOX2B$^+$/PRPH$^+$/ISL1$^+$/ISL2$^+$ and PHOX2B$^+$/TuJ1$^+$ neurons at 19 DDC (Fig. 6b). At 17 DDC, PHOX2B was downregulated in most NKX2.2$^+$ progenitors and cells expressing the 5HTN-lineage marker GATA3 had begun to emerge (Fig. 6b), indicating that progenitors underwent a MN-to-5HTN fate switch around this time. To reduce the amount of early-born MNs in culture, we introduced a dissociation step at 24 DCC as differentiated neurons showed lower capacity to reattach to the culture dish relative to immature precursors (Fig. 6a). Using this method, we could reduce the number of PHOX2B$^+$ MNs in culture at 34 DDC to ~6% (Fig. 6c, d) and attained cultures in which ~60–65% of total HuC/D$^+$ neurons expressed 5-HT and molecular markers characteristic of 5HTNs such as GATA3, LMX1B, tryptophan hydroxylase 2 (TPH2), serotonin transporter (SERT), and the 5-HT1A autoreceptor (5-HT1AR) (Fig. 6c–e). Together, these data provide proof-of-concept that timed duration of RA exposure can be applied for effective in vitro-derivation of human 5HTNs.

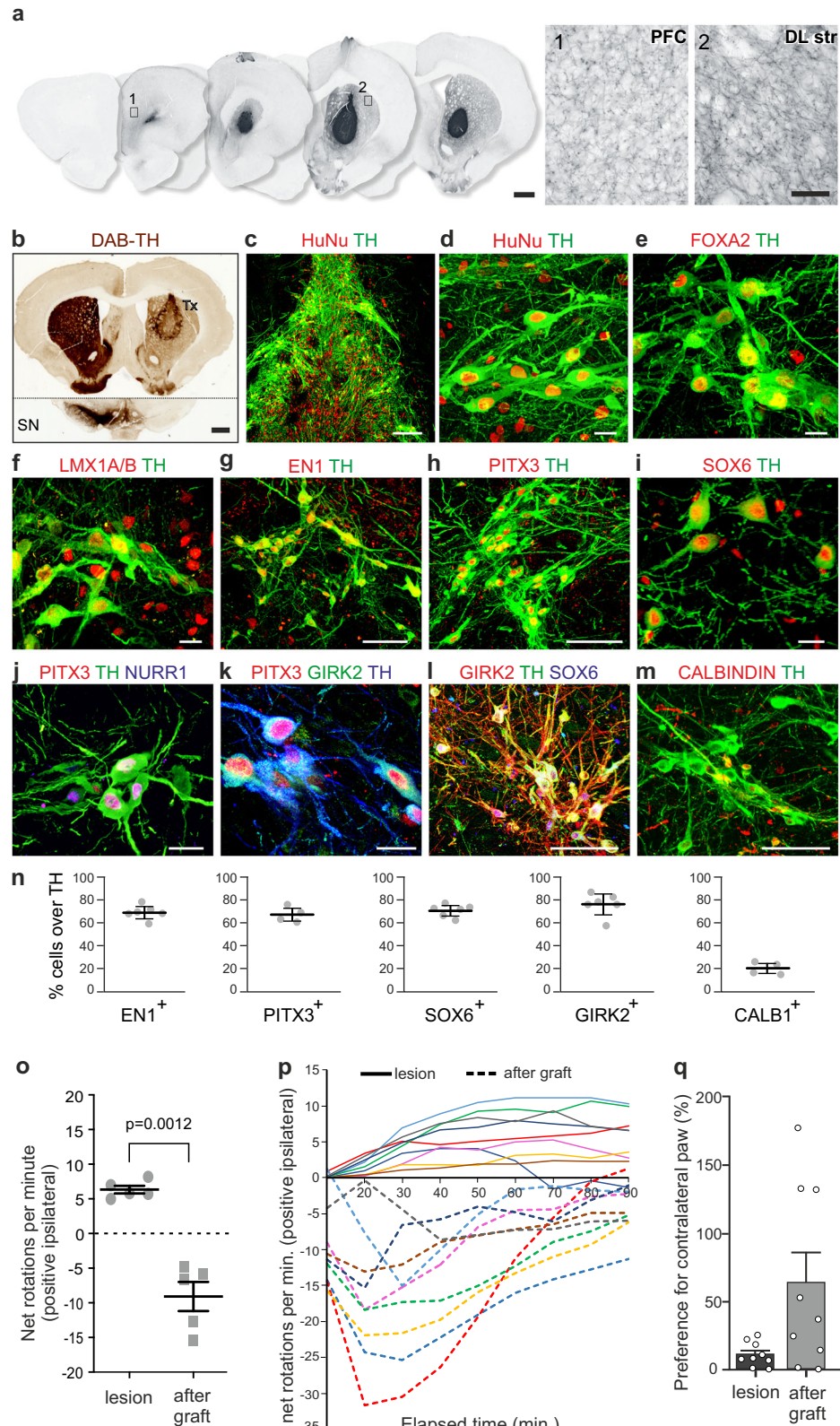

## Discussion

A central objective in stem cell research is the development of simple and robust differentiation techniques resulting in reproducible production of the desired cell product at high yield and purity[58]. In this study, we present an alternative approach for derivation of transplantable human mDA neurons which is principally different from existing state-of-the-art hPSC-based mDA neuron protocols, as it utilizes RA signaling instead of CHIR99021 or CHIR99021 and FGF8 to impose MB identity to hPSC-derived neural progenitors. When combined with SHH pathway activation, a 48-hour RA-pulse results in a consistent specification vMB progenitors with negligible contamination of cells expressing diencephalic or hindbrain identities. RA-specified vMB progenitors undergo progressive differentiation and maturation steps consistent with normal mDA neuron

**Fig. 5 RA-specified vMB preparations differentiate into functional dopaminergic neurons and restore motor deficits after transplantation into a rat model of PD. a–m** Immunohistological analysis of unilaterally 6-OHDA lesioned rats seven months after grafting of vMB preparations into the striatum. **a** Overview of graft-derived innervation to the host brain labeled with DAB-developed hNCAM staining, and magnification of human neuronal fiber outgrowth toward prefrontal cortex (PFC) and dorsolateral striatum (DLstr). **b** TH expression in striatum and substantia nigra (SN). Note TH immunoreactivity throughout the striatum and lack of TH expression in the SN on the lesioned side (right). **c–m** Immunohistochemistry of indicated markers in grafts. **n** Quantification of the percentage of TH$^+$ cells in grafts expressing EN1, SOX6, GIRK2 ($n = 6$ rats), PITX3 or CALB1 ($n = 4$ rats). Data presented as mean ± S.D. **o** Quantification of net rotations per minute in rats with baseline amphetamine-induced rotation scores ≥5 ipsilateral turns per minute ($n = 5$ rats) after lesion and seven months after transplantation. Data presented as mean ± S.E.M, $p$ value from Welch's unequal variance two-tailed $t$-test. **p** Rotational behavior over time after administration of amphetamine before (solid lines) and seven months after (dashed lines) grafting of all grafted animals ($n = 9$ rats). **q** Preference for contralateral paw use after lesion and seven months after transplantation. Data presented as mean±S.E.M ($n = 9$ rats). **o–q** the same rats analyzed post-lesion were analyzed seven months post-transplantation. Scale bars: 1000 μm in (**a**). left and (**b**), 100 μm in (**a**). right; 25 μm in (**d**, **e**, **f**, **I**, **j**, **k**); 10 μm in (**c**, **g**, **h**, **l**, **m**).

development resulting in high yield of mDA neurons expressing functional features in long-term cultures.

We show that RA is sufficient to impose progressively more caudal brain identities in a manner comparable to progressive activation of canonical WNT signaling by CHIR99021. However, a distinguishing feature with RA-based patterning is that regional specification is set by the duration of signal exposure and that the patterning response is relatively insensitive to altered RA concentrations. RA differs in this aspect to most developmental morphogen signals which specify different fates at different concentrations, including WNT[12,16]. We could assign the tolerance to variable levels of RA input to a negative feedback loop whereby fluctuations of RA input is counterbalanced by matching changes in rate of RA turnover by CYP26A1[43]. This regulatory circuitry is crucial for a reliable regional specification as the patterning response was disrupted after pharmacological inhibition of CYP26A1. Accordingly, these regulatory features provide robustness and reproducibility to the differentiation procedure and should contribute to reduce the need for batch-to-batch and line-to-line adjustments. This could facilitate differentiations where multiple hiPSC-lines are considered for clinical use or disease modelling in vitro[2,8], as well as scale-up efforts when a large number of cells are needed. Additionally, apart from promoting mDA neuron fate, we show that prolonged duration of RA exposure could be used to produce cranial MNs and 5HTNs indicating broad applicability of the differentiation paradigm. While our study firmly establish that RA is sufficient to specify mesencephalic fate independent of WNT and FGF8 signaling, a previous study has exploited a combination RA, WNT1, FGF8a, and SHH for hPSC-based derivation of mDA neurons[59]. In this study, a yield of ~1% FOXA2$^+$/TH$^+$ neurons was observed in long term cultures but FOXA2$^+$/TH$^+$ neurons were lost when WNT1 was selectively omitted from the differentiation procedure. The use of RA in this particular setting was consequently not sufficient to specify mDA neuron fate, and this could instead be allocated to WNT signaling.

We found that early RA exposure, in the context of dSMADi treatment, also promotes a fast and switch-like conversion of OCT4$^+$/SOX1$^-$ hPSCs into OCT4$^-$/SOX1$^+$ NSCs. This activity has a synchronizing effect on cells at the population level and presumably explains why RA-specified vMB progenitors undergo a relatively coordinated transition from a FP state to a neurogenic state ~2–3 weeks after RA treatment has ended. The RA-driven dissolution of pluripotency is also desirable from a safety perspective as it minimizes the potential risk for reactivation of pluripotency genes or that sporadic cells would maintain traits of pluripotency for more extended periods of time. However, with this said, it is important to stress that there are no reports of adverse side effect such as cellular overgrowth after extensive

preclinical evaluations or from initiated clinical experiments using preceding mDA neuron protocols[60,61].

When grafted into the striatum of 6-OHDA-lesioned rats, RA-specified vMB progenitors engraft, differentiate into mature mDA neurons that survive over long-term, and restore motor function. A majority of TH$^+$ neurons present in grafts expressed EN1 which is notably since EN1 is required for long-term survival of mDA neurons in mice[62,63]. Since our in vitro analyses establish negligible expression of EN1 in RA-specified vMB progenitors (Figs. 2e, 4m) we did not examine EN1 in the preparation used for grafting. We cannot, therefore, formally conclude that this batch lacked EN1 expression, which potentially could influence the numbers of EN1$^+$/TH$^+$ neurons generated. However, as upregulation of EN1 is observed in a subset of relatively young mDA neurons in vitro, it seems more plausible that the high proportion of EN1$^+$/TH$^+$ neurons in grafts reflects a progressive activation of EN1 in maturating RA-specified mDA neurons over time. Quantification of SOX6, PITX3, and GIRK2 in grafts further indicate that the major proportion of RA-specified mDA neurons express a A9-like subtype identity. Thus, RA-based vMB specification offers a potential alternative route to generate cells for cell therapy in PD. In this initial transplantation study we isolated vMB progenitors for grafting at a young FP-state which provided important proof-of-concept that mDA neurons generated using RA-based protocol function en par with mDA neurons generated using CHIR-based protocols[13,17,52,53].

As RA-specified cells undergo a relatively synchronized differentiation, it will in future experiments be exciting to explore if grafting of vMB progenitors isolated at other maturation stages potentially could improve yield, function, and innervation of mDA neurons, or the composition of the grafts[64]. The identification of RA as a MB patterning agent further provides an additional tool to explore if the combinatorial use of patterning signals can improve quality of preparations, as previously shown for the progressive development of CHIR99021/FGF8-based mDA neuron protocols[13,17,65]. For instance, recent CHIR-protocols utilized FGF8[17] or a second CHIR-boost[18] to reduce undesired diencephalic progenitors in preparations by promoting a caudal EN1$^+$ vMB progenitor identity, which improved graft outcome. RA-specified progenitors acquire a EN1$^-$ more rostral-like vMB identity without any significant contamination of diencephalic cells, which also resulted in a good graft outcome. In relation to this, it is notable that there is an uneven topological distribution of mDA neuron subtypes along the rostro-caudal axis of the adult MB, and with a concentration of A9-subtypes rostrally[51,66]. To what extent this relate to the positional origin of birth of neurons remains uncertain, but since RA imposes a rostral-like vMB identity while CHIR and FGF8 promotes more caudal-like vMB identities[17,18], it will be interesting to explore if sub-patterning of vMB progenitors by

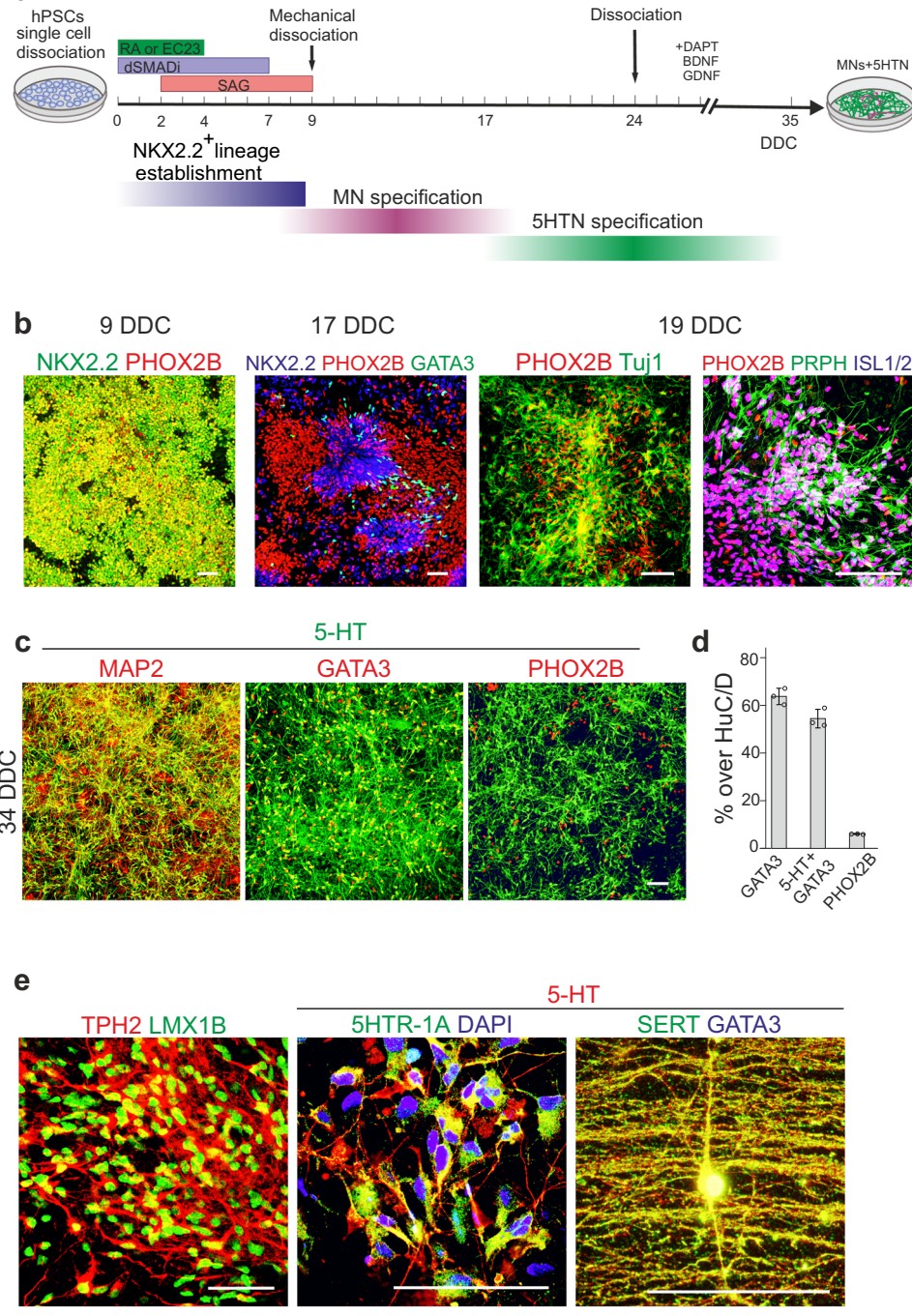

**Fig. 6 Specification of cranial MNs and 5HTNs. a** Schematic summary of differentiation conditions and processes timeline during MN and 5HTN generation. **b** Immunocytochemistry of NKX2.2, PHOX2B, and GATA3 at 9 and 17 DDC, and of Tuj1, PRPH, PHOX2B, and ISL1/2 at 19 DDC in cultures treated with RA[4D] + SAG. **c** Immunocytochemistry for serotonin (5-HT) together with MAP2, GATA3, or PHOX2B at 34 DDC. **d** Quantification of HuC/D[+] cells expressing PHOX2B, GATA3, or GATA3 and 5-HT, at 34 DDC (*n* = 3 independent differentiations). Data represented as mean ± S.D. **e** Immunocytochemistry of indicated markers in 5HTNs at 35-45 DDC. Scale bars: 100 μm in panels (**b**, **c**) and 50 μm in panel (**e**).

the combinatorial use of these signals can be used to modulate the subtype-composition of mDA neurons in grafts. Additionally, as RA acts upstream of signals emanating from the isthmic organizer, it is conceivable that sequential activation of RA and WNT signaling in hPSC-cultures more closely mimics normal embryonic development which could be beneficiary both regarding yield and functional efficiency of hPSC-derived mDA neurons after grafting.

## Methods

**Human PSC differentiation**. Human ESC lines (HS980 and HS401)[67] were provided by Drs. Outi Hovatta and Fredrik Lanner (Karolinska Institutet). Human iPSC lines (SM55 and SM56) were provided by Drs Pamela J. McLean and Simon Moussaud (Neuroregeneration Lab within the Center for Regenerative Medicine, Mayo Clinic). Isolation of primary human skin fibroblasts (obtained with patient consent) and generation of iPSCs were approved by the Mayo Clinic Institutional Review Board under IRB protocols (IRB 09-003803). Cells were maintained on recombinant human Vitronectin (VTN) (Thermo Fisher Scientific) coated plates in iPS-Brew XF medium (Miltenyi Biotech). Cells were passaged with EDTA

(0.5 mM), and ROCK inhibitor was added to the medium at a final 10 µM concentration for the first 24 h after plating. All cell lines tested negative for mycoplasma contamination. All differentiations were performed using hESC line HS980 unless indicated otherwise.

For PSC differentiation, 80–90% confluent PSCs cultures were rinsed twice with phosphate buffered saline (PBS), treated with EDTA (0.5 mM in PBS) for 5–7 min, and resuspended into single cell suspension in PBS. Cells were spin down at 400 g and resuspended in N2B27 medium (DMEM/F12: Neurobasal (1:1), 0.5 × N2 and 0.5 x B27 (plus vitamin A) supplements, 1 × nonessential amino acids, 1% GlutaMAX, 55 µM β-mercaptoethanol - all from Thermo Fisher Scientific) containing 5 µM SB431542 (Miltenyi Biotech) and 2.5 µM DMH1 (Santa Cruz Biotech) (dual SMAD inhibition), and 10 µM ROCK inhibitor (for the first 48 h after seeding). Cells were seeded on VTN (2 µg/cm²) and Fibronectin (FN) (2 µg/cm²) (Sigma) coated surface at a density of 60,000–80,000 cells/cm² for RA- and RA-analogues based experiments, and 20,000 cells/cm² for CHIR99021 experiments. SAG1.3 (Santa Cruz Biotechnology), CHIR99021 (Miltenyi Biotech), all-*trans* RA, 9-cis RA, 13-cis RA, Tazarotenic acid, R115866 (all Sigma), EC23 (Amsbio) were used at concentrations and time points described in the result section. All-trans RA and RA analogues were dissolved in DMSO (10 mM stock solution), aliquoted in light-protected condition and used within one month. Aliquots were stored at −20 °C and used once.

For mDA neuron differentiation (see schematic in Fig. 4) neural progenitors were mechanically dissociated at 9 DDC with Stem Cell Passaging Tool (Thermo Fisher Scientific) and seeded at 1:3 ratio in N2B27 medium containing 10 µM ROCK inhibitor (for first 48 h after dissociation) on VTN and FN coated surfaces. For terminal in vitro differentiation of dopaminergic neurons, cells were dissociated at 23 or 24 DDC with accutase (Thermo Fisher Scientific) and plated (600,000–800,000 cells/cm²) on VTN + FN + Laminin (2 µg/cm² each) (Sigma) coated surface in B27+ medium (Thermo Fisher Scientific) supplemented with BDNF (10 ng/ml) and GDNF (10 ng/ml) (Miltenyi Biotech), Ascorbic acid (0.2 mM) (Sigma), 10 µM ROCK inhibitor (Miltenyi Biotech) (for first 48 h after dissociation). For electrophysiology and neurotransmitter content analysis cells were grown in B27 Electrophysiology medium (Thermo Fisher Scientific) supplemented with BDNF (10 ng/ml) and GDNF (10 ng/ml) (Miltenyi Biotech), Ascorbic acid (0.2 mM) (Sigma) and 10 µM ROCK inhibitor (for first 48 h after dissociation) for at least 5 days before the experiment. The medium was routinely changed every 2–3 days.

For MN and 5HTN differentiation (see schematic drawing in Fig. 6) cells were resuspended in N2B27 medium containing 5 µM SB431542 and 2.5 µM DMH1, and 10 µM ROCK inhibitor (for the first 48 h after seeding) and seeded on VTN + FN (2 µg/cm² each) coated surface at a density of 60,000–80,000 cells/cm². Cells were treated with 200 nM of RA or EC23 for the first four days. 200 nM SAG1.3 was added to the differentiation media from 2–9 DDC. Cultures were mechanically dissociated at 9 DDC with Stem Cell Passaging Tool, resuspended in N2B27 medium containing 10 µM ROCK inhibitor (for first 48 hours after dissociation), and seeded at a 1:3 ratio on VTN + FN coated surfaces. For terminal in vitro differentiation of 5HTNs, cells were dissociated at 23 or 24 DDC with accutase and plated (600,000–800,000 cells/cm²) on VTN + FN + Laminin (2 µg/cm² each) coated surface in B27+ medium supplemented with 10 ng/ml BDNF, 10 ng/ml GDNF, 0.2 mM Ascorbic acid, 10 µM ROCK inhibitor (for first 48 h after dissociation), and 10 µM DAPT. The medium was routinely changed every 2–3 days.

**Immunocytochemistry and immunohistochemistry.** Cells were fixed for 12 min. at room temperature (RT) in 4% paraformaldehyde in PBS, rinsed three times in PBST (PBS with 0.1% Triton-X100), and blocked for 1 h at RT with blocking solution (3% FCS/0.1% Triton-X100 in PBS). Cells were then incubated with primary antibodies overnight at 4 °C, followed by incubation with fluorophore-conjugated secondary antibodies for 1 h at RT. Both primary and fluorophore-conjugated secondary antibodies were diluted in blocking solution. Appropriate Alexa (488, 555, 647)-conjugated secondary antibodies (Molecular Probes) were used. Immunohistochemistry was performed as described before [17]. Primary and secondary antibodies used are listed in Supplementary Table 4.

Immunofluorescence images were acquired with a confocal Zeiss LSM 700 microscope with ZEN 2011 software, Zeiss AxioImager M2 fluorescent microscope with ZEN2 software, and Molecular Devices ImageXpress Micro (software version 6.0), and analyzed with open-source ImageJ/Fiji (v1.53), CellProfiler (v3.1.5) or MetaXpress (v 5.0).

**Gene expression analysis.** Total RNA was isolated using Quick-RNA Mini Prep Plus kit (Zymo Research). cDNA was prepared using Maxima First Strand cDNA synthesis kit (Thermo Fisher Scientific). Quantitative Real-Time PCR was performed in a 7500 Fast Real Time PCR system thermal cycler using Fast SYBR Green PCR Master Mix (Applied Biosystems). Analysis of gene expression was performed using the $2^{-\Delta\Delta Ct}$ method, where relative gene expression was normalized to *GAPDH* transcript levels. Primers used are listed in Supplementary Table 3.

For Illumina RNA sequencing, RNA integrity was determined on an Agilent RNA 6000 Pico chip, using Agilent 2100 BioAnalyzer (Agilent Technologies). Illumina TruSeq Stranded mRNA kit with Poly-A selection was used for library construction. Samples were sequenced on NovaSeq6000 and HiSeq2500 instruments with a 2×51 setup. The Bcl to FastQ conversion was performed using bcl2fastq_v2.19.1.403 from the CASAVA software suite. Reads were mapped to the human genome assembly, build GRCh37 [https://www.ncbi.nlm.nih.gov/assembly/GCF_000001405.13/] using STAR (v 2.6.1a)[68]. Gene level abundances were estimated as FPKMs using StringTie2[69]. The read summarization to each gene (i.e. counts) was calculated using featureCounts[70]. Multiple ENSEMBL gene IDs were summed up per known HUGO gene symbol. The differential expression was estimated from log-transformed FPKM values. Statistical significance of differential expression was estimated with R function *t* test, using *p* values and false discovery rate (R package v. 3.6.2), assuming unequal variance and applying the Welsh degrees of freedom modification. Depending on the experimental setup, the design was considered either paired or unpaired. The resulting p-values of differential expression were accompanied with respective fold change values. The p-values were adjusted for multiple testing by calculating false discovery rate (FDR) by Benjamini and Hochberg's method[71].

Heatmap plotting and PCA visualization were performed with online tools (beta-version) at [https://www.evinet.org/][72] using standard parameter settings of R package heatmaplyand function princompas back end.

**Western blot.** Cells were lysed in RIPA buffer (Sigma) complemented with protease and phosphatase inhibitor cocktail (ThermoFisher Scientific) and incubated on ice with shaking for 30 min. Lysate was cleared by centrifugation (20,000 g for 20 min at 4 °C) and protein concentration was determined by Bicinchoninic Acid (BCA) assay. Protein lysate was resuspended in LDS buffer (Thermo Fisher Scientific) containing 2.5% 2-Mercaptoethanol and denaturated at 95 °C for 5 min. 15–30 µg of protein were loaded per lane of a 4–15% SDS polyacrylamide gel (Bio-Rad) and transferred onto nitrocellulose membranes (BioRad) using Trans-Blot Turbo System (BioRad). Membranes were incubated in blocking solution (TBS with 0.1% Tween-20 (TBST) and 5% nonfat dry milk) for 1 h at RT, followed by overnight incubation at 4 °C with primary antibodies. After 3 washes with TBST at RT, membranes were incubated with HRP- conjugated secondary antibodies for 1 h at RT. Detection of HRP was performed by chemiluminescent substrate Super-Signal West Dura substrate and the signal was detected on a ChemiDoc Imaging System (Bio-Rad). Primary antibodies for immunoblotting are listed in listed in Supplementary Table 4.

**HPLC.** Concentrations of noradrenaline (NA), dopamine (DA), and serotonin (5-HT) in 42 DDC cultures were determined by high-performance liquid chromatography (HPLC) with electrochemical detection.

Cultures at 42 DDC differentiated in dSMADi+RA2D were incubated in physiological solution (140 mM NaCl; 2.5 mM KCl; 1 mM MgCl₂; 1.8 mM CaCl₂; buffered with HEPES (20 mM) at pH 7.4) or in high K⁺ solution (physiological solution with 56 mM K⁺, and the concentration of Na⁺ ions was proportionally reduced to keep the same total osmolarity). After 20 min. (min.), incubation solutions were collected and used for analysis of neurotransmitter content. To determine cellular neurotransmitter content, cells were lysed in H₂O. Incubation solutions and cellular lysates were deproteinized with 0.1 M perchloric acid and after 15 min. incubation on ice, samples were double centrifuged at 20,000 g for 15 min. as described before[73]. Protein concentration was determined by BCA method. Samples were then analyzed in a HPLC system consisting of HTEC500 (Eicom, Kyoto, Japan) and a CMA/200 Refrigerated Microsampler (CMA Microdialysis, Stockholm, Sweden) equipped with a 20 µl loop and operating at +4 °C. The potential of the glassy carbon working electrode was + 450 mV vs. the Ag/AgCl reference electrode. Separation was achieved on a 200 × 2.0 mm Eicompak CAX column (Eicom). The mobile phase was a mixture of methanol and 0.1 M phosphate buffer (pH6.0) (30:70, v/v) containing 40 mM potassium chloride and 0.13 mM EDTA-2Na. The chromatograms were recorded and integrated using the computerized data acquisition system Clarity (DataApex).

**Quantification of neurite length and neuron branching.** Average neurite length was quantified using NeuronJ plugin in ImageJ package[74] [https://imagej.net/plugins/neuronj] on immunofluorescent images of TH⁺ neurons (magnification 10x, 0.5 zoom). The number of branches per neuron was calculated using immunofluorescent images of TH⁺ neurons at higher magnification, 20x.

**Electrophysiology.** Slides containing 35–60 days old neurons were placed in a recording chamber containing electrophysiology medium (Neurabasal Medium, Electro; Thermo Fisher Scientific). For recording, neurons were visualized using a DIC microscope (Scientifica, Uckfield, UK) with a 60x objective (Olympus, Tokyo, Japan). Patch pipettes (resistance 3–5 MΩ for voltage clamp recordings, 5–10 MΩ for current clamp recordings), pulled on a P-87 Flaming/Brown micropipette puller (Sutter Instruments, Novato, CA, USA), were filled with either 154 mM NaCl solution for voltage clamp recordings or 120 mM KCl solution containing 8 mM biocytin for current clamp recordings. Signals were recorded with an Axon MultiCalmp 700B amplifier and digitized at 20 kHz with an Axon Digidata 1550B digitizer (Molecular Devices, San Jose, CA, USA). Access resistance and pipette capacitance were compensated. Cell attached voltage clamp recordings were bandpass filtered at 2 Hz low/1 kHz high and events showing an after hyperpolarization

were considered spontaneous action potentials. To assess spiking patterns, neurons recorded in current clamp mode were held at a membrane potential of −60 mV. Near-threshold current steps were applied to determine the rheobase current, then 1 s current steps proportional to the rheobase current were applied. The presence of $Na^+$ and $K^+$ currents was determined in voltage clamp mode by applying a voltage step with intervals of 10 mV from a holding of −60 mV. Currents were measured from baseline (10 ms before the voltage step onset) to the peak of the characteristic fast $Na^+$ current component or the peak of the slow $K^+$ current component. The electrical properties were extracted using custom written Matlab (MathWorks, Natick, MA, USA) script [https://zenodo.org/record/6367837#.YjSC3DUo9PY]. After recording, slides were fixed, stained, and imaged as described above. Biocytin was visualized using Streptavidin Alexa Fluor 488 (Thermo Fisher Scientific).

**Graft placement and behavioral analysis**. All animal procedures were performed in accordance with the European Union Directive (2010/63/EU) and were approved by the Malmö/Lund Ethics Committee for the use of laboratory animals (Malmö/Lunds djurförsöksetiska nämnd) and the Swedish Department of Agriculture (Jordbruksverket). Adult female, athymic "nude" rats were purchased from Harlan/Envigo Laboratories (Hsd:RH-Foxn1rnu) and were housed with ad libitum access to food and water, under a 12-hr light/dark cycle. Animals were at a minimum of 16 weeks at start of experiment (180 g). Surgical procedures and lesion of the nigrostriatal pathway by a single unilateral injection of 6-hydroxydopamine (6-OHDA) into the right medial forebrain bundle (MFB) were performed as described previously[75]. Briefly, surgical procedures were performed under general anesthesia using a solution of fentanyl and medetomidine (20:1) injected intraperitoneally (1 mL/kg; Apoteksbolaget, Sweden). Lesion of the nigrostriatal pathway in "nude" rats was induced by unilateral injection of 6-hydroxydopamine into the right MFB, with a volume of 4 µL at freebase concentration of 3.5 mg/mL to the following coordinates relative to bregma: A/P −4; M/L −1.2; D/V (from dura) −7.5. Lesion severity was measured four weeks after 6-OHDA injection by amphetamine-induced rotations recorded over 90 min using an automated system (Omnitech Electronics)[17] and by stepping test[13]. Amphetamine-induced rotation was induced by intraperitoneal injection of 2.5 mg/kg amphetamine hydrochloride (Sigma). Eight weeks after 6-OHDA MFB lesion, "nude" rats received a total dose of 150,000 hESC-derived progenitors at day 14 of differentiation into the striatum as previously described[75]. Briefly, a volume of 2 µL of cell suspension (cells at a concentration 75,000 cells/µL were injected at a rate of 1 µL per minute followed by a diffusion time of 2 minutes) was injected to the striatum at the following coordinates relative to bregma: A/P + 0.5; M/L −3; D/V (from dura) −4.5; adjusted to flat head. Amphetamine-induced rotation and stepping test were assessed 7 months after grafting. Animals were perfused after behavioral analysis and then processed for immunohistochemistry.

**Statistical analysis**. $P$ values were calculated using Student's $t$-test, Welch's unequal variance $t$-test or pairwise Wilcoxon test and were performed for statistical analysis of two groups. Statistical methods used are indicated in the corresponding figure legends.

**Reporting summary**. Further information on research design is available in the Nature Research Reporting Summary linked to this article.

## Data availability
Source Data for Figs. 1–6 and Supplementary Figs. 1–7 are provided with this paper. RNA-seq datasets reported in this manuscript have been deposited to the Gene Expression Omnibus under accession number GSE147404. Human genome assembly, build GRCh37 [https://www.ncbi.nlm.nih.gov/assembly/GCF_000001405.13/].

## Code availability
Script and data for analysis of neuron electrical properties is available at [https://zenodo.org/record/6367837#.YjSC3DUo9PY].

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

## Acknowledgements

We thank Drs. Outi Hovatta and Fredrik Lanner (Karolinska Institutet) for hESC lines, Drs. Pamela J. McLean and Simon Moussaud, Neuroregeneration Lab within the Center for Regenerative Medicine, Mayo Clinic (Mayo Clinic, Jacksonville) for hiPSC lines. We acknowledge support from the National Genomics Infrastructure in Genomics Production, Stockholm for assistance with massively parallel sequencing and access to the UPPMAX computational infrastructure. This work was supported by grants from Knut and Alice Wallenberg Foundation (KAW2011.0661, KAW2012.0101; JE), Swedish Research Council (2013-4155, 2017-02089; JE), The Swedish Foundation for Strategic Research (SRL10-0030; JE), Hjärnfonden (FO2017-0037, F02019-0154, F02021-0159; JE), Parkinson Fonden (1189/19, 1257/20, 1326/21; JE), Novo Nordisk Foundation (NNF20OC0062355; JE), Karolinska Institutet (JE).

## Author contributions

Z.A.: conception and study design, hPSC differentiation and characterization, data analysis and interpretation. J.M.D and M.K.: hPSC manipulation, differentiation and characterization, data interpretation. A.F.A.: in vivo transplantation, behavioral analyses, transplantation data analysis with contribution from S.N. J.A.v.L. and M.C.: electrophysiology and data analysis. A.J. and A.A.: RNA-seq data analysis and bioinformatics. S.V.: hPSC differentiation and characterization. J.K. and T.Y.: HPLC analysis. M.P. design and supervision of transplantation studies. J.E.: supervision, conception and study design, and writing of manuscript together with Z.A, J.M.D., A.F.A., and M.P.

## Funding

## Competing interests

A patent application (no. 2006792.2) pertaining to the results presented in the paper has been filed by JE and ZA with support from Karolinska Institute Innovations AB. M.P. is the owner of Parmar Cells AB and co-inventor of the following patents WO2016162747A2, WO2018206798A1 and WO2019016113A1. The remaining authors declare no competing interests.

**Additional information**

