## [Peer Review File · Nature Communications]

Robust derivation of transplantable dopamine neurons from human pluripotent stem cells by timed retinoic acid deliveryREVIEWER COMMENTS

Reviewer #1 (Remarks to the Author):

The study by Alekseenko et al. describes a new approach to the efficient generation of ventral midbrain dopaminergic progenitor cells from hPSCs through the use of timed RA stimulation in lieu of concentration-dependent WNT activation by GSK3 inhibition, as has been applied in several previously published protocols. The study very convincingly shows that the timing of RA addition to the cultures very accurately controls the caudalisation of hPSCs into either midbrain fates (2 days of RA) or hindbrain fates (3-4 days of RA). This result on patterning is remarkably similar to the effect of increasing doses of GSK3i, with the main difference that RA-mediated patterning is sensitive to time whereas GSK3i-mediated patterning is sensitive to concentration. The authors claim that the insensitivity to the RA concentration makes the protocol more robust and easy to transfer between cell lines. This is an important point of value to the field, but the claim must be substantiated further by bath-to-batch variation analyses. Furthermore, a main concern of the study is the peculiar lack of EN1 expression in the RA-patterned VM progenitors – and neurons(?). This point must be addressed with further analyses

My specific comments are:

- Multiple studies have shown that EN1 is a marker of the caudal ventral midbrain, where the SNpc DA neurons arise, and that EN1 is expressed in DA progenitors and neurons and is required for the long-term survival of the mature DA neurons (Veenvliet et al., Dev 2013; Alberi et al., Dev 2004; Simon et al., J. Neuroscience 2001; Nouri/Awatrami, Dev 2017; Kee et al., Cell Stem Cell 2017). For this purpose, other hPSC protocols specifically aim to induce the expression of EN1 in the ventral midbrain progenitors by adding FGF8 or high WNT signalling to the cells to simulate MHB signalling in later stages of the protocol; after midbrain patterning has taken place (i.e. Wan Kim/Studer et al., Cell Stem Cell 2021; Nolbrant et al., 2017). The protocol presented in this study does not include any MHB signalling molecules to induce caudalisation of the VM progenitors, and as such it is perhaps not surprising that the cultures do not express EN1. However, it leaves behind the question as to which subtype of DA neurons can arise from a cell which is OTX2+/LMX1+/FOXA2+/EN1-? Some specific questions related to this arise:

1. It is claimed that “a small fraction” of the DA neurons start to express EN1 upon terminal maturation, but it is unclear to what degree this takes place. To uncover this, the expression of EN1 should be followed in the cultures by qRT-PCR over time (from progenitors until terminally mature neurons) and the number of EN1+/TH+ neurons in the cultures should further be quantified by ICC to clarify how much EN1 is induced in the cultures during neuronal maturation

2. Similarly, it is shown through a single image in Fig. 5h that some neurons in the graft are also EN1-positive, but the extent of EN1 expression in the graft is unclear. More low-magnification images of EN1 expression as well as quantification of EN1+/TH+ neurons through random sampling of images in the graft should be performed. In addition, it is important to clarify if the batch of cells which was used for transplantation had the same characteristics as the other batches analysed in vitro to rule out that the transplanted cells may have been differently patterned or contaminated with EN1-expressing cells of other origins (i.e. hindbrain or roof plate). Please provide RNA or ICC data specifically of the transplanted batch of cells.

3 It is claimed in the abstract that the RA-based protocol is of advantage compared to other protocols, as it results in less batch-to-batch variation, and it can robustly be transferred to other cell lines. These claims are not sufficiently substantiated by data in the manuscript. In particular:

4. An analysis of batch-to-batch variation has not been performed in the manuscript apart from two batches being analysed in parallel in Fig. 2c. In order to truly claim robustness across batches, at least 6 batches of cells from the main hESC line with SAG + RA-2D should be compared in parallel through the full panel of qRT-PCR markers used in the study.

5. Similarly, since the robustness of the protocol and the easy transfer to other cell lines is a main selling point of the study, the batch-to-batch variation should also be assessed in three independent biological replicate differentiations of each of the 4 cell lines used in the study (Fig. 3a). ICC images for LMX1A, FOXA2, OTX2 and DAPI for each of these batches should be provided (i.e. could be suppl fig) and/or a panel of qRT-PCR markers can be used for assessment.

6 In the results section, the authors claim that the TH+ neurons derived in vitro are positive for DAT and VMAT2. (Fig. 4I). However, maturation of DA neurons in 2D culture to a state where robust DAT and VMAT can be detected is notoriously difficult, and the commercially available antibodies are often hampered by unspecific background when applied on human cells. It is noteworthy that the staining for DAT and VMAT2 in Fig. 4I seems to label many cells in the dish that are not TH-positive, even though these markers are supposed to be restricted only to DA neurons. To determine if the stainings with these antibodies are specific, the authors should perform a control staining of other non-DA human neuronal culture in parallel to the DA neuron, and this data should be presented in the study (i.e. suppl material). If the ICC for DAT and VMAT turns out to be hampered by unspecific background staining, then this data should be removed from the manuscript.

- Minor comments:

1. Please state in the beginning of the results section which cell line is being used for the studies.
2. RA is known to be a highly unstable compound which is rapidly degraded upon exposure to light and atmospheric air. Since reproducibility is a key component of the study, the authors should specify in the methods section exactly how the RA was handled in order to ensure reproducibility, i.e. which solvent was used for the RA, how was it stored, how many times did the aliquots undergo free-thaw cycles, and were any other precautions taken to limit degradation of the RA?
3. Please specify the seeding density used for the terminal maturation of the neurons (i.e. bottom of page 17)
4. Please specify in Fig. 5 (figure or legend) at which time point after transplantation the rotational responses were recorded

Reviewer #2 (Remarks to the Author):

This MS by Alekseenko et al reports a novel protocol for generating midbrain dopamine neurons (mDA) from human ESCs and iPSCs. Currently the most widely used protocol for mDA differentiation of hPSCs is based on Wnt activation by a GSK3b inhibitor CHIR. First in man clinical trials are underway and at least one of the trials is based on the CHIR strategy. Instead of CHIR, the new protocol by Alekseenko et al use RA as a patterning cue. The authors show convincingly that following a 48 hour RA exposure during neural induction period, cells with gene expression characteristic of mDA neural progenitors were induced. This was demonstrated by RT-PCR, antibody staining and bulk RNAseq. Gene/marker expression indicative of a caudal diencephalon identity was low at transcript level and cells expressing protein markers of this region were a minority. The RA patterned neural progenitor cultures later produce a high proportion of TH+ dopamine neurons that co-express midbrain markers FOXA2 and LMX1B, as well as some other pan-dopaminergic markers. The authors provide evidence that the RA patterning activity is modulated by self-enhanced RA degradation via CYP26A1. Moreover, a pilot transplantation of RA patterned progenitors was performed in nud rats that showed good post graft differentiation/survival of mDA neurons and an improvement in amphetamine induced rotation.

The authors argue that their protocol is robust and, unlike the CHIR protocol that may require careful titration of CHIR for individual hPSC lines and basal culture media used, RA patterning is

determined by duration of treatment(48 hours) rather than a precise concentration, hence this new method could be used as an alternative method for cell therapy and disease modeling. The MS is well written and data nicely presented.

Specific comments:

1, Robustness of the protocol. While the data presented demonstrate a high mDA inductive efficiency, its reproducibility between independent differentiation runs of the same hPSC line, and crucially in different PSC lines is not clear. There is no information whether the key quantitative data re. mDA identity/efficiency and the presence of caudal diencephalic fate (Fig 2h, Fig 4g, 4i) were from a single line or else. Since this is described as the highlight of this new protocol, quantitative data on mDA induction at the progenitor and postmitotic neuron stage from each of the four ESC/iPSC lines should be provided, so the readers could truly appreciate the advance of this protocol. Data on the generation of caudal diencephalic cells from different lines/independent experiments is of particular important.

2, The authors suggest the presence of A9 mDA in their culture based on GIRK2 expression. This may be misleading as embryonic GIRK2 expression is not a reliable readout for A9 fate, and published scRNAseq data don't support A9/A10 segregation or their presence in human fetal midbrain cells and hESC-derived mDA cultures.

3, The authors describe their new method as principally different from that using CHIR. It would be interesting to learn whether their cells truly experience distinct environment during the course of differentiation. Were WNT and/or FGF pathway activated at all based on the RNAseq analysis?

Response to the Reviewers

We thank the reviewers for their critical and constructive assessment of our work. In the following, we address each specific point raised by the referees.

Reviewer # 1 (Remarks to the Author):

The study by Alekseenko et al. describes a new approach to the efficient generation of ventral midbrain dopaminergic progenitor cells from hPSCs through the use of timed RA stimulation in lieu of concentration-dependent WNT activation by GSK3 inhibition, as has been applied in several previously published protocols. The study very convincingly shows that the timing of RA addition to the cultures very accurately controls the caudalisation of hPSCs into either midbrain fates (2 days of RA) or hindbrain fates (3-4 days of RA). This result on patterning is remarkably similar to the effect of increasing doses of GSK3i, with the main difference that RA-mediated patterning is sensitive to time whereas GSK3i-mediated patterning is sensitive to concentration. The authors claim that the insensitivity to the RA concentration makes the protocol more robust and easy to transfer between cell lines. This is an important point of value to the field, but the claim must be substantiated further by bath-to-batch variation analyses. Furthermore, a main concern of the study is the peculiar lack of EN1 expression in the RA-patterned VM progenitors – and neurons(?). This point must be addressed with further analyses

My specific comments are:

- Multiple studies have shown that EN1 is a marker of the caudal ventral midbrain, where the SNpc DA neurons arise, and that EN1 is expressed in DA progenitors and neurons and is required for the long-term survival of the mature DA neurons (Veenvliet et al., Dev 2013; Alberi et al., Dev 2004; Simon et al., J. Neuroscience 2001; Nouri/Awatrami, Dev 2017; Kee et al., Cell Stem Cell 2017). For this purpose, other hPSC protocols specifically aim to induce the expression of EN1 in the ventral midbrain progenitors by adding FGF8 or high WNT signalling to the cells to simulate MHB signalling in later stages of the protocol; after midbrain patterning has taken place (i.e. Wan Kim/Studer et al., Cell Stem Cell 2021; Nolbrant et al., 2017). The protocol presented in this study does not include any MHB signalling molecules to induce caudalisation of the VM progenitors, and as such it is perhaps not surprising that the cultures do not express EN1. However, it leaves behind the question as to which subtype of DA neurons can arise from a cell which is OTX2+/LMX1+/FOXA2+/EN1-? Some specific questions related to this arise:

To address the subtype identity mDA neurons generated from RA-specified OTX2+/LMX1+/FOXA2+/EN1⁻ vMB progenitors, we have in our revised manuscript extended our immunocytochemical analysis of TH⁺ neurons in grafts (Revised Fig.5c-m) and quantified the proportion of TH⁺ neurons that express EN1, PITX3, SOX6, GIRK2 and CALB1 (revised Fig.5n). This analysis reveals that ~70% of mDA neurons express detectable EN1 after grafting, showing that a majority of RA-specified differentiating mDA neurons upregulated EN1 over time (revised Fig. 5n) despite that EN1 not is expressed at progenitor stages (Fig.2e and revised Fig.4m). This could help to explain why RA-specified mDA neurons survive over long-term which is clearly the case, albeit it should be noted that a requirement of EN1 for long-term survival of mDA neurons has been established in mouse and not human cells.

Importantly, SOX6 and GIRK2 are molecular markers enriched in A9-type mDA neurons and we find that ~70-80% of TH⁺ neurons in grafts co-express these markers (revised Fig. 5n). Conversely, ~20% of TH⁺ neurons express CALB1 (revised Fig. 5n) which is enriched in A10-type mDA neurons. Accordingly, these new data and quantifications suggest that a majority RA-specified OTX2⁺/LMX1⁺/FOXA2⁺/EN1⁻ vMB progenitors differentiate into a A9-like subtype and a minor fraction into a A10-like subtype after grafting.

EN1 is expressed in a caudal-high to rostral-low gradient in the vMB progenitor domain, with essentially undetectable expression levels rostrally (dos Santos and Smidt, Neural Dev, 2011). Since RA-specified vMB progenitors express a OTX2⁺/LMX1⁺/FOXA2⁺/EN1⁻ identity, we therefore define these as “rostral-like”. The relationship between mDA neuron subtypes and rostro-caudal origin of birth remain poorly resolved, and there is to our knowledge no lineage-tracing experiments arguing that A9-subtypes (SNpc) would be exclusively derived from caudal EN1^{HIGH} vMB progenitors. By contrast, topological analyses show a clear enrichment of A9-subtypes (lateral and dorsal tier) in the rostral MB while A10-subtypes instead are concentrated caudally. A9 subtypes (lateral tier) are also present caudally (see Brignani and Pasterkamp, Front Neuroanat.,2017; Smits et al., Plos One 2013; Veevliet et al, Dev 2013). Rostro-caudal origin of birth of neurons may contribute to this topology, in particular since A10 subtypes show very limited migration after leaving the ventricular zone at birth. If so, rostral EN1^{-LOW} vMB progenitors would be predicted to generate a high proportion of A9 subtypes, which indeed is consistent with our complemented in vivo analysis suggesting that RA-specified OTX2⁺/LMX1⁺/FOXA2⁺/EN1⁻ vMB progenitors generate a high proportion of A9-like neurons after grafting.

Also, albeit the newly developed CHIR-based protocols that the referee refers to utilize different caudalizing strategies, the main purpose for these were to improve reproducibility of vMB patterning, and not to induce caudal EN1^{HIGH} vMB identity with the intention of increasing the fraction of A9-subtypes. The Parmar laboratory applied FGF8 to caudalize cells in order to reduce the fraction of contaminating diencephalic progenitors in preparations (Kirkeby et al., Cell Stem Cell, 2017). Similarly, in their recent Cell Stem Cell paper (Kim et al, Cell Stem Cell, 2021), Studer and co-workers disclose that their original and widely used CHIR-based protocol (Kriks, Nature, 2011) shows limitations regarding robustness and reproducibility. With their new CHIR-boost, they attain more consistent vMB patterning and they provide evidence that induction of EN1, in this protocol, contribute to suppress diencephalic contamination.

Since the precise relation between mDA subtypes and rostro-caudal origin of birth is still unclear, we avoid any extensive discussion on this topic. However, since RA imposes a rostral-like character while CHIR and FGF8 induce a more caudal-like character, we discuss the potential to explore if the combinatorial use these signals can be used to sub-pattern vMB progenitors and thereby modulate the proportion of mDA neuron subtypes generated. In the discussion of the revised manuscript on page 17 we write: *“There is also an uneven topological distribution of mDA neuron subtypes along the rostro-caudal axis of the adult MB, and with a concentration of A9-subtypes rostrally^{50,63}. To what extent this relate to the positional origin of birth of neurons remains uncertain, but since RA imposes a rostral-like vMB identity while CHIR and FGF8 promotes more caudal-like vMB identities^{17,64}, it will be interesting to explore if sub-patterning of vMB progenitors by the combinatorial use of these signals can be used to modulate the subtype-composition of mDA neuron in grafts.”*

1. It is claimed that “a small fraction” of the DA neurons start to express EN1 upon terminal maturation, but it is unclear to what degree this takes place. To uncover this, the expression of EN1 should be followed in the cultures by qRT-PCR over time (from progenitors until terminally mature neurons) and the number of EN1+/TH+ neurons in the cultures should further be quantified by ICC to clarify how much EN1 is induced in the cultures during neuronal maturation

In the revised manuscript, we have analyzed expression of *EN1* over time by qPCR. Consistent with RNA-seq data included in the original version of the manuscript (Fig.2e), these new data show negligible *EN1* expression at early differentiation stages while a notable upregulation of *EN1* is observed in hESC-cultures at 40DDC. These data are included in Fig. 4m of the revised manuscript. As requested, we have also quantified EN1 expression in vitro, which shows that ~15% of TH⁺ neurons express detectable EN1 in long-term cultures. We present this new data in Fig. 4l,n in the revised manuscript. Thus, while EN1 is not expressed at progenitor stages (as also emphasized in the original version of the manuscript), these data indicate that EN1 becomes upregulated in RA-specified differentiating mDA neurons over time. In addition, and as an important support for this (and as requested by the referee in point 2), quantifications show that a majority of TH⁺ neurons (~70%) co-express EN1 in grafts (see below).

2. Similarly, it is shown through a single image in Fig. 5h that some neurons in the graft are also EN1-positive, but the extent of EN1 expression in the graft is unclear. More low-magnification images of EN1 expression as well as quantification of EN1+/TH+ neurons through random sampling of images in the graft should be performed. In addition, it is important to clarify if the batch of cells which was used for transplantation had the same characteristics as the other batches analysed in vitro to rule out that the transplanted cells may have been differently patterned or contaminated with EN1-expressing cells of other origins (i.e. hindbrain or roof plate). Please provide RNA or ICC data specifically of the transplanted batch of cells.

As requested, we have replaced the EN1/TH immunostaining in the revised manuscript with a lower magnification micrograph (Fig. 5g). We have also quantified the proportion of TH⁺ neurons that co-express EN1. Quantitative data from grafts in 6 individual rats show that ~70% of TH⁺ mDA neurons express EN1 in grafts seven months after transplantation. These new data are shown in Fig.5n in the revised manuscript.

In our grafting experiment in rats, we confirmed a high proportion of LMX1A⁺ and FOXA2⁺ cells in preparations. We did not examine EN1 as numerous preceding differentiations had firmly established that EN1 was not detectable in RA-specified vMB preparations, as addressed in the results section in the original and revised manuscript. Unfortunately, we do not have cDNA from the preparation used and cannot therefore examine *EN1* expression in retrospect. However, we find it highly unlikely that the vMB preparation used for transplantation would differ significantly from other vMB differentiations, in particular since we have data showing that TH⁺ neurons in grafts express EN1 and several additional mDA neuron markers including FOXA2, NURR1, LMX1A/B, PITX3, SOX6 and GIRK2. Due to their strict vMB-origin, one would not have expected mDA neuron-rich grafts if the RA-specified preparation used for grafting would have been inappropriately patterned or contained contaminating EN1⁺ cells from a non-vMB origin. We cannot provide EN1 expression data for the preparation used for grafting, but we hope our reasoning regarding this issue will be satisfactory for the referee.

3 It is claimed in the abstract that the RA-based protocol is of advantage compared to other protocols, as it results in less batch-to-batch variation, and it can robustly be transferred to other cell lines. These claims are not sufficiently substantiated by data in the manuscript. In particular:

4. An analysis of batch-to-batch variation has not been performed in the manuscript apart from two batches being analysed in parallel in Fig. 2c. In order to truly claim robustness across batches, at least 6 batches of cells from the main hESC line with SAG + RA-2D should be compared in parallel through the full panel of qRT-PCR markers used in the study.

In the revised manuscript, we have included qPCR analyses of six biological replicates for our standard hESC-line HS980 differentiated towards a vMB fate by RA^{2D}+SAG-treatment. Cells were isolated for analysis at day 14 and the expression of 17 regional markers were examined by qPCR. As reference cell types, we included two forebrain (FB)-differentiations (No RA+SAG) and two hindbrain (HB)-differentiations (RA^{4D}+SAG). Collectively, these analyses support a low inter-experimental variability and consistent vMB-specification by RA, with low expression of inappropriate FB- or HB-markers. These new data are presented in Fig.2h and referred to in the results section on page 8 of the revised manuscript: *“qPCR analyses of six biological replicates of RA^{2D}+SAG cultures isolated at 14DDC indicated consistent vMB specification and low expression of inappropriate regional markers (Fig.2h).”*

5. Similarly, since the robustness of the protocol and the easy transfer to other cell lines is a main selling point of the study, the batch-to-batch variation should also be assessed in three independent biological replicate differentiations of each of the 4 cell lines used in the study (Fig. 3a). ICC images for LMX1A, FOXA2, OTX2 and DAPI for each of these batches should be provided (i.e. could be suppl fig) and/or a panel of qRT-PCR markers can be used for assessment.

In the revised manuscript, we have included quantitative data for LMX1A, FOXA2 and NKX2.1 for three biological replicates for each hPSC-line in Fig.2i and primary data for the expression of LMX1A, FOXA2, OTX2, NKX2.1 and DAPI for the 3 replicates for lines hESC HS401, hiPSC SM56 and hiPSC SM55 in Supplementary Fig.3. In summary, these analyses reveal a comparable induction LMX1A, FOXA2 and OTX2 at high yield for each cell line, without need to adjust the differentiation procedure. Additionally, we also observed low number of NKX2.1⁺ cells in all cell lines analyzed, indicating minimal presence of contaminating diencephalic progenitors.

We believe these new data significantly improved the quality of our study, and we have made the following modifications to the text in the results section on page 9 of the revised manuscript: *“Next we compared the patterning response to RA^{2D}+SAG treatment for one additional hESC-line HS 401 as well as two hiPSC-lines SM55 and SM56. For each cell line analyzed at 14DDC, we observed consistent induction of LMX1A⁺/FOXA2⁺/OTX2⁺/NKX2.1⁻ vMB progenitors at high yield without need to adjust concentration or time of RA exposure (Fig.2i and Supplementary Fig.3), which would normally require re-titering of patterning agents using other mDA neuron protocols where CHIR is used for caudalization¹⁵. Collectively, these data suggest that RA-based differentiation promotes robust and reproducible vMB specification with high inter-experimental consistency and low cell line variability.”*

6 In the results section, the authors claim that the TH+ neurons derived in vitro are positive for DAT and VMAT2. (Fig. 4l). However, maturation of DA neurons in 2D culture to a state where robust DAT and VMAT2 can be detected is notoriously difficult, and the commercially available antibodies are often hampered by unspecific background when applied on human cells. It is noteworthy that the staining for DAT and VMAT2 in Fig. 4l seems to label many cells in the dish that are not TH-positive, even though these markers are supposed to be restricted only to DA neurons. To determine if the stainings with these antibodies are specific, the authors should perform a control staining of other non-DA human neuronal culture in parallel to the DA neuron, and this data should be presented in the study (i.e. suppl material). If the ICC for DAT and VMAT2 turns out to be hampered by unspecific background staining, then this data should be removed from the manuscript.

We thank the referee for this information as we were not aware that commercial DAT and VMAT2 antibodies can be problematic and show unspecific immunoreactivity on human cells. As the referee suggested, we have examined these antibodies on hPSC-derived PHOX2B⁺ cranial motor neurons and found indeed that the VMAT2 antibody used shows unspecific staining. We have therefore removed the VMAT2 ICC data from the revised manuscript and have instead included qPCR analyses of differentiating hPSC-cultures (9-40 DDC) showing induction of VMAT2 (*SLC18A2*) expression at the mRNA level at 40DDC. This new data is presented in Fig. 4m. The DAT antibody did not show any significant immunoreactivity on cranial motor neurons, and we have therefore kept the DAT ICC in Fig. 4l in the revised manuscript. We also include complementary data in Fig. 4m showing that *DAT* (*SLC6A3*) is expressed at the mRNA level and included DAT ICC on cranial motor neuron cultures in Supplementary Fig. 5d.

- Minor comments:

1. Please state in the beginning of the results section which cell line is being used for the studies.

We have included this information in the begin of the results section on page 5, and in the Methods section on page 18 we have also included the following sentence: *“All differentiations were performed using hESC line HS980 unless indicated otherwise.”*

2. RA is known to be a highly unstable compound which is rapidly degraded upon exposure to light and atmospheric air. Since reproducibility is a key component of the study, the authors should specify in the methods section exactly how the RA was handled in order to ensure reproducibility, i.e. which solvent was used for the RA, how was it stored, how many times did the aliquots undergo free-thaw cycles, and were any other precautions taken to limit degradation of the RA?

We agree with the reviewer on the importance of how RA is handled, and we have introduced this information in the methods section on page 18: *“All-trans RA and RA analogues were dissolved in DMSO (10mM stock solution), aliquoted in light-protected condition and used within one month. Aliquots were stored at -20°C and used once.”*

3. Please specify the seeding density used for the terminal maturation of the neurons (i.e. bottom of page 17)

We have introduced in the revised manuscript cell seeding concentration used for terminal differentiations in the description of the differentiation protocol in the methods section on page 19.

4. Please specify in Fig. 5 (figure or legend) at which time point after transplantation the rotational responses were recorded

The rotational responses we performed 7 months after transplantation of vMB cell preparations. We now clearly state this in the figure legend of Figure 5.

Reviewer #2 (Remarks to the Author):

This MS by Alekseenko et al reports a novel protocol for generating midbrain dopamine neurons (mDA) from human ESCs and iPSCs. Currently the most widely used protocol for mDA differentiation of hPSCs is based on Wnt activation by a GSK3b inhibitor CHIR. First in man clinical trials are underway and at least one of the trials is based on the CHIR strategy.

Instead of CHIR, the new protocol by Alekseenko et al use RA as a patterning cue. The authors show convincingly that following a 48 hour RA exposure during neural induction period, cells with gene expression characteristic of mDA neural progenitors were induced. This was demonstrated by RT-PCR, antibody staining and bulk RNAseq. Gene/marker expression indicative of a caudal diencephalon identity was low at transcript level and cells expressing protein markers of this region were a minority. The RA patterned neural progenitor cultures later produce a high proportion of TH+ dopamine neurons that co-express midbrain markers FOXA2 and LMX1B, as well as some other pan-dopaminergic markers. The authors provide evidence that the RA patterning activity is modulated by self-enhanced RA degradation via CYP26A1. Moreover, a pilot transplantation of RA patterned progenitors was performed in nud rats that showed good post graft differentiation/survival of mDA neurons and an improvement in amphetamine induced rotation.

The authors argue that their protocol is robust and, unlike the CHIR protocol that may require careful titration of CHIR for individual hPSC lines and basal culture media used, RA patterning is determined by duration of treatment(48 hours) rather than a precise concentration, hence this new method could be used as an alternative method for cell therapy and disease modeling.

The MS is well written and data nicely presented.

Specific comments:

1, Robustness of the protocol. While the data presented demonstrate a high mDA inductive efficiency, its reproducibility between independent differentiation runs of the same hPSC line, and crucially in different PSC lines is not clear. There is no information whether the key quantitative data re. mDA identity/efficiency and the presence of caudal diencephalic fate (Fig 2h, Fig 4g, 4i) were from a single line or else. Since this is described as the highlight of this new protocol, quantitative data on mDA induction at the progenitor and postmitotic neuron stage from each of the four ESC/iPSC lines should be provided, so the readers could truly appreciate the advance of this protocol. Data on the generation of caudal diencephalic cells from different lines/independent experiments is of particular important.

Data in Fig.2h and Fig. 4g,i in the original manuscript were generated using our standard hESC-line HS980. In the revised manuscript, we now state in the very beginning of the results section on page 5 that we use GMP-compliant hESC-line HS980 to make this clearer. Additionally, in the Methods section on page 18 we have also included the following sentence: *“All differentiations were performed using hESC line HS980 unless indicated otherwise.”*

To strengthen data on robustness, we have in the revised manuscript included qPCR analyses of six biological replicates for our standard hESC-line HS980 differentiated towards a vMB fate by RA^{2D}+SAG-treatment. Cells were isolated for analysis at day 14 and the expression of 17 regional markers were examined by qPCR. As reference cell types, we included two forebrain (FB)-differentiations (No RA+SAG) and two hindbrain (HB)-differentiations (RA^{4D}+SAG). These analyses support a low inter-experimental variability and consistent vMB-specification, with low expression of

inappropriate FB- or HB-markers These new data are presented in Fig.2h and referred to in the results section on page 8 of the revised manuscript: “qPCR analyses of six biological replicates of RA^{2D}+SAG cultures isolated at 14DDC indicated consistent vMB specification and low expression of inappropriate regional markers (Fig.2h).”

In line with the referee’s request, we have in the revised manuscript included quantitative data on LMX1A, FOXA2 and NKX2.1 expression in progenitor cells at 14 DDC for three biological replicates for each of the four hPSC-lines included in our study. These data are presented in Fig.2i of the revised manuscript. We have also included in Supplementary Fig.3 primary data for the expression of LMX1A, FOXA2, OTX2, NKX2.1 and DAPI for the 3 replicates for cell lines hESC HS401, hiPSC SM56 and hiPSC SM55. In summary, these analyses reveal a comparable induction of LMX1A, FOXA2 and OTX2 at high yield for each cell line, without need to adjust the differentiation procedure. Additionally, we also observed low number of NKX2.1⁺ cells in all cell lines analyzed, indicating minimal presence of contaminating diencephalic progenitors.

We have also extended the characterization of mDA neuron differentiation to the additional cell lines (hESC HS401, hiPSC SM56 and hiPSC SM55) and quantified the percentage of TH⁺ neurons in culture and expression of LMX1B in TH⁺ cells, which show a high yield of TH⁺ mDA neurons in all four lines analyzed (~60-80%). This data is now included in the revised manuscript and presented in Supplementary Fig.5c.

We believe these new data significantly improved the quality of our study, and we have made the following modifications to the text in the results section. On page 9 of the revised manuscript we included: “Next we compared the patterning response to RA^{2D}+SAG treatment for one additional hESC-line HS 401 as well as two hiPSC-lines SM55 and SM56. For each cell line analyzed at 14DDC, we observed consistent induction of LMX1A⁺/FOXA2⁺/OTX2⁺/NKX2.1⁻ vMB progenitors at high yield without need to adjust concentration or time of RA exposure (Fig.2i and Supplementary Fig.3), which would normally require re-titering of patterning agents using other mDA neuron protocols where CHIR is used for caudalization¹⁵. Collectively, these data suggest that RA-based differentiation promotes robust and reproducible vMB specification with high inter-experimental consistency and low cell line variability.”. On page 12 of the revised manuscript we have introduced: “All four hPSC-lines examined in this study were confirmed to give rise to a high content of TH⁺ neurons (~60-80% of total neurons) in long-term cultures (Supplementary Fig. 5c).”

2, The authors suggest the presence of A9 mDA in their culture based on GIRK2 expression. This may be misleading as embryonic GIRK2 expression is not a reliable readout for A9 fate, and published scRNAseq data don’t support A9/A10 segregation or their presence in human fetal midbrain cells and hESC-derived mDA cultures.

We agree with the referee that expression of GIRK2 and CALB1 in young differentiating mDA neurons is not conclusive regarding mDA neuron subtypes, and we have therefore removed the statement about GIRK2 and CALB1 as markers enriched in A9 and A10 subtypes, respectively, from the result section addressing the in vitro characterization of mDA neurons. In this section we instead state more generically on page 12: “Cells also expressed the mDA neuron maturation markers DAT, GIRK2, SYNAPTOPHYSIN, CALB1, SOX6, VMAT2 and ALDH1A1 at 40 DDC, as determined by immunocytochemistry (Fig. 4l and Supplementary Fig. 5b,d) or mRNA expression (Fig.4m).”

In the revised manuscript, we instead raise the issue about mDA neuron subtypes in the section of the results addressing graft outcome in rats, which is more appropriate as RA-specified mDA neurons in grafts are 7.5 months old. In the revised manuscript, we have extended our immunohistochemical expression analyses (presented in the revised Fig.5c-m) and quantified the proportion of TH⁺ neurons that express EN1, PITX3, SOX6, GIRK2 and CALB1 (presented in revised Fig.5n). Importantly, SOX6 and GIRK2 are markers enriched in A9-type mDA neurons (Brignani and Pasterkamp, Front Neuroanat. ,2017) and we show that ~70-80% of TH⁺ neurons in grafts co-express these markers (new Fig. 5n). Conversely, ~20% of TH⁺ neurons in grafts express the A10-enriched marker CALB1 (new Fig. 5m,n). These complementary analyses provide important data suggesting that a majority of RA-specified mDA neurons (≥70%) adopt a A9-like molecular identity after differentiation and maturation in vivo. In the results section of the revised manuscript on page 14, we adjusted the text accordingly: “≥ 70% of TH⁺ neurons also co-expressed PITX3 as well as GIRK2 and SOX6, which are markers enriched in therapeutic A9-subtype of mDA neurons⁵⁰ (Fig.5i-l,n). The A9 identity was further supported by TH⁺ fibers innervating the surrounding dorsolateral striatum (Fig.5a and Supplementary Fig.7c). Innervation was also detected in prefrontal cortex and a small fraction of TH⁺ neurons expressed CALB1 (20.5% ± 4.7, mean± SD) (Fig.5m,n), a marker enriched in A10 subtypes of mDA neurons⁵⁰”. In the discussion we have therefore added a sentence on page 17: “Quantification of SOX6, PITX3, and GIRK2 in grafts further indicate that the major proportion of RA-specified mDA neurons express a A9-like subtype identity.”

3, The authors describe their new method as principally different from that using CHIR. It would be interesting to learn whether their cells truly experience distinct environment during the course of differentiation. Were WNT and/or FGF pathway activated at all based on the RNAseq analysis?

We do believe that we addressed that RA appears to specify a LMX1A⁺/FOXA2⁺/OTX2⁺ vMB identity independent of WNT1 or FGF8 induction. In our original manuscript, we show in Fig.2e that *WNT1* and *FGF8* are not expressed in vMB preparations at 14DDC and that RA, in contrast to CHIR, does not promote nuclear accumulation of β-catenin in nuclei (Fig.2f). In the revised manuscript, we have also included data on *WNT1*, *FGF8* and the *WNT* target gene *AXIN2* in ESC-cultures between 2-14DDC (new data included in Supplementary Fig.2e). In the results section of the revised manuscript on page 9, we address this information and suggest that RA appears to induce vMB identity independently of WNT1 or FGF8: “There was no nuclear accumulation of β-catenin in response to RA treatment (Fig.2f) and no induction of *WNT1* or the *WNT* response gene *AXIN2*, nor *FGF8* in response to RA in ESC-cultures at 2DDC to 14DDC (Fig.2e and Supplementary Fig.2e) Together, this suggest that LMX1A⁺/FOXA2⁺/OTX2⁺ cells induced by RA and SAG acquire a EN1^{-LOW} rostral-like vMB identity and that specification of vMB-fate can occur independently of WNT1, FGF8 or induction of isthmic organizer-like cells in hPSC-cultures.”

REVIEWER COMMENTS

Reviewer #1 (Remarks to the Author):

In the revised version of the manuscript, the authors have added additional evidence for the robustness of the presented differentiation protocol as well as quantification of EN1 expression in the grafts. This is an improvement, and the data is well presented, showing that the mature DA neurons in the grafts indeed to express EN1 to a high degree. However, the data in the manuscript is still in conflict with several previous studies on the requirement for EN1 expression in the immature mDA progenitor cells. The authors have not been able to clarify whether the single transplanted batch of cells was representative of other in vitro batches with regards to the expression of EN1, and it is peculiar that the cells in the grafts showed much higher EN1 levels (70% EN1+) than cells from other batches matured in vitro (15% EN1). Although the authors claim that EN1 can be induced over time to some extent in vitro (although still at low levels compared to other protocols), it cannot be concluded from the present data whether the relatively high EN1 expression in the grafted cells was obtained from cultures with a low or high initial expression of EN1 in the progenitors. If no additional data can be presented to clarify this inconsistency with previous studies, then a thorough discussion must be included in the Discussion section on the lack of detectable EN1 expression in the progenitor cells of the current protocol and how this is in conflict with previous studies showing that EN1 is expressed in both immature DA progenitor cells and DA neurons; that it is required for DA neuron differentiation and that it is required for generating for DA-rich grafts from stem cells (Veenvliet et al., Dev 2013; Alberi et al., Dev 2004; Simon et al., J. Neuroscience 2001; Nouri/Awatrami, Dev 2017; Kee et al., Cell Stem Cell 2017, Kirkeby et al., Cell Stem Cell 2017; Wan Kim/Studer et al., Cell Stem Cell 2021). In line with this, it should also be clearly stated in the discussion section that the level of EN1 expression in the progenitor cells of the transplanted batch was not determined and that it could not be ruled out that this batch of cells could potentially have had a higher level of EN1 expression at the time of transplantation compared to other batches.

An additional comment to p4, line 82-84: It should be correctly noted here for ref 16 and 17 that it was specifically the caudal vMB fates which were found to correlate to grafts with high dopaminergic content, and that a main purpose of the FGF8 and CHIR-boost treatments was to induce the expression of EN1 in the progenitors for obtaining correct DA -specification in the grafts. It should be noted in the text also that Ref 18 is not a published study, but a pre-print. Several previous studies have also shown that CHIR-based patterning is highly concentration sensitive, so these might also be referenced.

Reviewer #2 (Remarks to the Author):

The authors adequately addressed my comments.

REVIEWER COMMENTS

Reviewer #1 (Remarks to the Author):

In the revised version of the manuscript, the authors have added additional evidence for the robustness of the presented differentiation protocol as well as quantification of EN1 expression in the grafts. This is an improvement, and the data is well presented, showing that the mature DA neurons in the grafts indeed to express EN1 to a high degree. However, the data in the manuscript is still in conflict with several previous studies on the requirement for EN1 expression in the immature mDA progenitor cells.

As addressed in our initial letter of response to the referees, we do not agree that the data in our study conflicts with any studies arguing for a requirement for EN1 expression in immature mDA progenitors. In particular, *in vivo* studies show that EN1 expression is confined only to the caudal half of the MB in mice at embryonic day 9.5 (Hynes and Rosenthal, *Current Opinion in Neurobiology*, 1999; Liu and Joiner, *Annu. Rev. Neurosci.* 2001; dos Santos and Smidt, *Neural Dev*, 2011) and rostrally derived mDA neurons are consequently derived from vMB progenitors that do not express EN1 *in vivo*. Based on this, it is not very surprising that mDA neurons can be generated also from hPSC-derived EN1⁻ vMB progenitors *in vitro*, which is what we observe using RA-based vMB patterning.

We are not aware of any study arguing for an absolute requirement for EN1 progenitor expression for the generation of mDA neurons. We assume the referee refers to studies from the Parmar and Studer labs on improved mDA neuron protocols, but we do not believe that these conflict with our data. In these studies, they utilize a second CHIR-boost (Studer) or FGF8 (Parmar) to impose a caudal EN1⁺ vMB identity, which results in a more reproducible vMB specification and improved graft outcome (less variable) relative to earlier CHIR-based protocols. However, as outlined below, these studies do not reveal a requirement for EN1 expression in progenitors for the generation of mDA neurons. As extensively discussed in Kirkeby et al., *Cell Stem Cell*, 2017, the primary rationale to apply FGF8 to caudalize progenitors was to eliminate/reduce contaminating diencephalic progenitors in preparations. By doing so, they increased the fraction of vMB progenitors with potential to produce mDA neurons and achieved a more consistent graft outcome. Albeit they identify EN1 as a positive prognostic marker for good graft outcome, there is no data arguing that expression of EN1 progenitors is required for mDA neuron generation. Similarly, in Kim et al., *Cell Stem Cell*, 2021, the Studer lab shows that a second CHIR-boost caudalizes progenitors into a EN1⁺ vMB identity and that diencephalic gene expression is reduced, resulting in more reproducible vMB specification. In this study they do inactivate EN1 in hPSCs and show that this results in increased diencephalic gene expression. This indicates that induction of EN1 contributes to suppress diencephalic genes, in their protocol. However, expression of EN1 in progenitors is not required for mDA neuron generation *per se*. Their mRNA expression analyses at 30 days show a reduction, but not a loss, of mDA neuron markers *NURR1* and *PITX3* and this is accompanied by an increase in expression of subthalamic markers. Accordingly, mDA neurons are generated in EN1^{-/-} hPSC-cultures, but presumably in lower numbers since a more significant fraction of cells select a diencephalic fate of differentiation in the absence of EN1 function.

In our study we show that RA-based patterning results in induction of EN1⁻ rostral-like LMX1A⁺/FOXA2⁺/OTX2⁺ vMB progenitors, and with negligible contamination of undesired diencephalic progenitors (even though EN1 not is expressed). Accordingly, when RA is used as vMB

patterning agent, additional signaling cues are not needed to suppress diencephalic fate. In this condition, RA-specified rostral-like LMX1A⁺/FOXA2⁺/OTX2⁺/EN1⁻ vMB progenitors effectively differentiate into mDA neurons in vitro and result in mDA neuron-rich grafts after transplantation in vivo. As argued above, we do not understand why these findings would conflict with previous studies on CHIR-based vMB specification. Nevertheless, in line with the referee's request, we have in the discussion of the revised manuscript added the following sentences in the final paragraph of the discussion on page 18 (new text is marked in yellow): *"The identification of RA as a MB patterning agent further provides a new tool to explore if the combinatorial use of patterning signals can improve quality of preparations, as previously shown for the progressive development of CHIR99021/FGF8-based mDA neuron protocols^{13,16,65}. For instance, recent CHIR- protocols utilized FGF8¹⁷ or a second CHIR-boost¹⁸ to reduce undesired diencephalic progenitors in preparations by promoting a caudal EN1⁺ vMB progenitor identity, which improved graft outcome. RA-specified progenitors acquire a EN1⁻ more rostral-like vMB identity without any significant contamination of diencephalic cells, which also resulted in a good graft outcome."*

The authors have not been able to clarify whether the single transplanted batch of cells was representative of other in vitro batches with regards to the expression of EN1, and it is peculiar that the cells in the grafts showed much higher EN1 levels (70% EN1+) than cells from other batches matured in vitro (15% EN1). Although the authors claim that EN1 can be induced over time to some extent in vitro (although still at low levels compared to other protocols), it cannot be concluded from the present data whether the relatively high EN1 expression in the grafted cells was obtained from cultures with a low or high initial expression of EN1 in the progenitors.

We don't believe it is peculiar that the number of EN1 positive neurons differ in vitro and in vivo, since mDA neurons analyzed in vitro are young (~2-3weeks after birth from immature progenitors) while neurons in grafts have differentiated and matured over a period of 7 months. Therefore, we believe the most plausible explanation for the high number of EN1⁺/TH⁺ neurons in grafts reflects a progressive activation of EN1 in maturing RA-specified mDA neurons over time. We have never observed any significant expression of EN1 in RA-specified vMB progenitors in vitro, and we did not therefore consider examining EN1 in the batch used for grafting. We therefore agree with the referee that we formally cannot exclude that the batch used for grafting was completely devoid of EN1 expression. In line with the referees request, we have added the following text in the discussion on page 17: *"A majority of TH⁺ neurons present in grafts expressed EN1 which is notably since EN1 is required for long-term survival of mDA neurons in mice^{62,63}. Since our in vitro analyses establish negligible expression of EN1 in RA-specified vMB progenitors (Fig. 2e,4m) we did not examine EN1 in the preparation used for grafting. We cannot, therefore, formally conclude that this batch lacked EN1 expression, which potentially could influence the numbers of EN1⁺/TH⁺ neurons generated. However, as upregulation of EN1 is observed in a subset of relatively young mDA neurons in vitro, it seems more plausible that the high proportion of EN1⁺/TH⁺ neurons in grafts reflects a progressive activation of EN1 in maturing RA-specified mDA neurons over time."*

If no additional data can be presented to clarify this inconsistency with previous studies, then a thorough discussion must be included in the Discussion section on the lack of detectable EN1 expression in the progenitor cells of the current protocol and how this is in conflict with previous studies showing that EN1 is expressed in both immature DA progenitor cells and DA neurons; that it is required for DA neuron differentiation and that it is required for generating for DA-rich grafts from stem cells (Veenvliet et al., Dev 2013; Alberi et al., Dev 2004; Simon et al., J. Neuroscience 2001;

Nouri/Awatrami, Dev 2017; Kee et al., Cell Stem Cell 2017, Kirkeby et al., Cell Stem Cell 2017; Wan Kim/Studer et al., Cell Stem Cell 2021).

As addressed in detail above, we added a statement in the discussion on page 18 in which we aim to briefly explain why both CHIR-boost- and CHIR/FGF8-specified EN1⁺ vMB progenitors and RA-specified EN⁻ vMB progenitors can all generate mDA neuron-rich grafts: *“For instance, recent CHIR-protocols utilized FGF8¹⁷ or a second CHIR-boost¹⁸ to reduce undesired diencephalic progenitors in preparations by promoting a caudal EN1⁺ vMB progenitor identity, which improved graft outcome. RA-specified progenitors acquire a EN1⁻ more rostral-like vMB identity without any significant contamination of diencephalic cells, which also resulted in a good graft outcome.”*

In line with this, it should also be clearly stated in the discussion section that the level of EN1 expression in the progenitor cells of the transplanted batch was not determined and that it could not be ruled out that this batch of cells could potentially have had a higher level of EN1 expression at the time of transplantation compared to other batches.

As indicated above, we have now added the requested statement in the discussion on page 17 of the revised manuscript: *“Since our in vitro analyses establish negligible expression of EN1 in RA-specified vMB progenitors (Fig. 2e,4m) we did not examine EN1 in the preparation used for grafting. We cannot, therefore, formally conclude that this batch lacked EN1 expression, which potentially could influence the numbers of EN1⁺/TH⁺ neurons generated. However, as upregulation of EN1 is observed in a subset of relatively young mDA neurons in vitro, it seems more plausible that the high proportion of EN1⁺/TH⁺ neurons in grafts reflects a progressive activation of EN1 in maturing RA-specified mDA neurons over time.”*

An additional comment to p4, line 82-84: It should be correctly noted here for ref 16 and 17 that it was specifically the caudal vMB fates which were found to correlate to grafts with high dopaminergic content, and that a main purpose of the FGF8 and CHIR-boost treatments was to induce the expression of EN1 in the progenitors for obtaining correct DA -specification in the grafts.

We have in the revised manuscript modified the sentence in accordance with the referee’s request, and also included the associated information that these new methods reduce diencephalic contamination. On page 4 we now state: *“Assessment of a large set of grafting experiments have further linked inter-experimental variability to imprecise vMB specification and contamination of diencephalic progenitors in spite of optimized CHIR-titration, which could be adjusted for by the complementing caudalizing activity of FGF8¹⁶ or with biphasic CHIR-treatment (CHIR-boost)¹⁷ which reduced diencephalic contamination and induced a more caudal EN1⁺ vMB identity of cells.”*

It should be noted in the text also that Ref 18 is not a published study, but a pre-print. Several previous studies have also shown that CHIR-based patterning is highly concentration sensitive, so these might also be referenced.

In the revised manuscript, we now indicate that reference 18 is a pre-print.

We agree with the referee that previous published studies show that CHIR-based patterning is concentration sensitive, and we refer to this in the previous version of the manuscript on page 4, where we stated: *“...the anterior-posterior (AP) patterning response of differentiating hPSCs to CHIR is concentration-sensitive which requires careful protocol adjustments for some cell lines^{14,15}.”* To make this point clearer we have added two additional references and modified the sentence: *“...the anterior-posterior (AP) patterning response of differentiating hPSCs to CHIR is highly concentration-sensitive and requires careful protocol adjustments between cell lines^{12,14-16}.”*

By modifying the text and discussion in accordance with the referee's requests, we now hope the referee will find our manuscript suitable for publication in Nature Communications.

Reviewer #2 (Remarks to the Author):

The authors adequately addressed my comments.

REVIEWERS' COMMENTS

Reviewer #1 (Remarks to the Author):

The comments have been adequately addressed

REVIEWER COMMENTS

Reviewer #1 (Remarks to the Author):

The comments have been adequately addressed

Reviewer #2 (Remarks to the Author):

The authors adequately addressed my comments.

We are pleased to have been able to address all the comments by reviewers.